# Review of Reinforcement Learning for Large Language Models: Formulations, Algorithms, and Opportunities

## Abstract

Large Language Models (LLMs) represent significant milestones in artificial intelligence development. While pre-training on vast text corpora and subsequent supervised fine-tuning establish their core abilities, Reinforcement Learning (RL) has emerged as an indispensable paradigm for refining LLMs, particularly in aligning them with human values, and teaching them to reason and follow complex instructions. As this field evolves rapidly, this survey offers a systematic review of RL methods for LLMs, with a focus on fundamental concepts, formal problem settings, and the main algorithms adapted to this context. Our review critically examines the inherent computational and algorithmic challenges arising from the integration of RL with LLMs, such as scalability issues, effective gradient estimation, and training efficiency. Concurrently, we highlight exciting opportunities for advancing LLM capabilities through new RL strategies, including multi-modal integration and the development of agentic LLM systems.

## 1 Introduction

Large Language Models (LLMs) have experienced rapid development in recent years, transforming from academic research tools to powerful systems with widespread commercial applications. The evolution from early models like GPT-1 (Radford et al., 2018) to current state-of-the-art systems such as GPT-5 (OpenAI, 2025), Claude-4 (Anthropic, 2025), Gemini-2.5 (Comanici et al., 2025), Llama-4 (Meta, 2025), Qwen-3 (Yang et al., 2025a), and DeepSeek-R1 (Guo et al., 2025) not only reflects exponential growth in model size, capabilities, and adoption but also signals a new era for AI. This acceleration has been driven by several key factors: architectural innovations in transformer models (Vaswani et al., 2017; Li et al., 2025d), increasingly large and diverse training datasets (Brown et al., 2020), and significant advancements in computational resources dedicated to training (Kaplan et al., 2020; Hoffmann et al., 2022).

The development of modern LLMs typically involves several distinct phases: initial pre-training on vast text corpora (Radford et al., 2019; Brown et al., 2020), refinement through Supervised Fine-Tuning (SFT) on specialized datasets (Raffel et al., 2020), and crucially, alignment and reasoning enhancement through Reinforcement Learning (RL) techniques (Christiano et al., 2017; Ouyang et al., 2022; Guo et al., 2025). Each developmental stage addresses specific limitations and enhances particular model capabilities: pre-training establishes foundational knowledge and linguistic understanding; SFT teaches instruction following and appropriate response formatting; and RL is instrumental in boosting alignment, safety, and complex reasoning abilities. These post-training techniques are proving indispensable for strong downstream task performance.

The transformative power of post-training methodologies, particularly those involving RL, becomes evident through compelling empirical results. According to OpenAI's technical report (OpenAI, 2023), the GPT-4 base model only marginally outperformed GPT-3.5 before post-training optimization. However, after applying Reinforcement Learning from Human Feedback (RLHF), GPT-4 achieved a remarkable 30-point improvement on TruthfulQA (Lin et al., 2021)—a benchmark specifically designed to evaluate a model's ability to discern facts from adversarially-selected falsehoods. Similarly, on competition-level mathematics problems from AIME 2024, OpenAI's o1 model (OpenAI, 2024)—trained with specialized RL techniques to enhance reasoning capabilities—achieved an impressive 83.3% score. This marks an absolute improvement of approximately 70

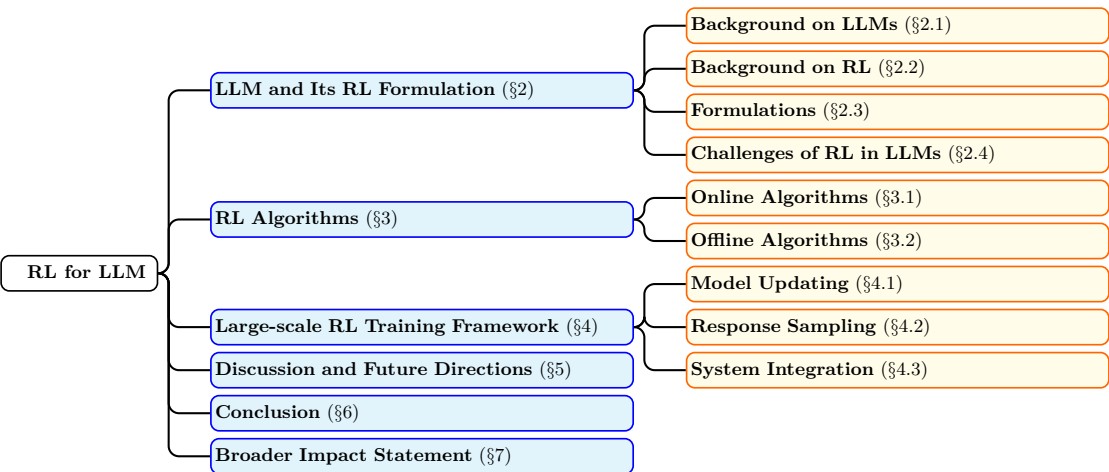

Figure 1: The organization of survey.

percentage points over their previous flagship, GPT-4o, which scored around 13.3% on the same problems. These dramatic performance improvements underscore the critical importance of post-training approaches, particularly those leveraging RL methodologies.

The application of RL in LLM development has expanded rapidly across diverse domains and objectives. Key areas include value alignment tasks that ensure models behave according to human preferences (Ouyang et al., 2022; Bai et al., 2022a;b; Ji et al., 2023), safety enhancement to reduce harmful outputs (Dai et al., 2023; Guan et al., 2024; Beutel et al., 2024), specialized domain optimization such as mathematical reasoning and code generation (Roziere et al., 2023; Shao et al., 2024; Hui et al., 2024), and emerging agentic capabilities that enable models to perform complex, multi-step tasks (Yao et al., 2023a; Yang et al., 2024b; Zheng et al., 2025d). Each application domain presents unique challenges in reward design, training stability, and evaluation metrics.

These expanding applications have driven significant innovations across multiple dimensions: new problem formulations that better capture the nuances of language-based tasks (Wei et al., 2022b; Rafailov et al., 2023), algorithmic advances that address the unique challenges of RL in high-dimensional spaces (Li et al., 2024c; Shao et al., 2024), and sophisticated training infrastructures capable of handling the computational demands of large-scale RL training (Yao et al., 2023b; Sheng et al., 2024). The rapid pace of development in this intersection has created a pressing need for systematic analysis and consolidation of emerging methodologies.

Driven by the rapid advancements in Reinforcement Learning (RL) and its pivotal role in enhancing modern Large Language Model (LLM) capabilities, this survey provides a systematic analysis of the intersection between RL and LLMs, with a particular focus on recent methodological advances. The organization of this survey is detailed in Figure 1:

- **Fundamental RL Concepts and Formulations:** We introduce core RL concepts (Sutton, 1988) and examine how they have been adapted for language model training. Special attention is given to the unique characteristics of RL in the LLM context, including reward modeling in text-based environments, challenges of estimating gradients, and performing gradient updates in the high-dimensional space.

- **Leading RL Training Algorithms**: We provide a comprehensive analysis of prominent RL algorithms that have demonstrated success in LLM training, including PPO (Schulman et al., 2017), ReMax (Li et al., 2024c), and GRPO (Shao et al., 2024). Our examination highlights the connections and distinctions between these algorithms and conventional RL methods, adaptations required for language models' unique characteristics, and trade-offs between computational efficiency and learning effectiveness.

- **Distributed RL Training Frameworks:** We review existing infrastructures and methodologies for implementing RL at the scale required for modern LLMs, discussing memory-efficient training

techniques, accelerated generation techniques (Kwon et al., 2023), strategies for distributed training and optimization (Sheng et al., 2024), and approaches to handle the computational challenges of large-scale RL deployment (Wu et al., 2025).

- **Future Directions and Open Challenges:** We outline promising research avenues at the RL-LLM intersection. This includes exploring necessary modifications to current methodologies, emerging techniques for more sample-efficient learning, strategies to address current limitations in scalability and stability, and advancing LLM capabilities through multi-modal integration and the development of agentic LLM systems.

We additionally provide a dedicated broader-impact discussion (Section 7) covering safety risks (e.g., reward hacking), value pluralism ("whose preferences?"), bias amplification in reward optimization, capability trade-offs (alignment tax), and the energy footprint of large-scale RL.

**Scope and relation to existing surveys.** To maintain focus, this survey deliberately concentrates on the methodological intersection of RL and LLM post-training. It therefore does not attempt to comprehensively cover pre-training methodologies (Albalak et al., 2024), model explainability (Zhao et al., 2024), or concrete domain-specific applications of LLMs (Casper et al., 2023; Wang et al., 2025a). Several recent surveys and overviews study adjacent aspects of this area: general alignment surveys provide broad taxonomies of AI alignment risks, objectives, and evaluation protocols (Ji et al., 2023); reward-design-oriented surveys emphasize how feedback signals, reward models, and safety constraints shape LLM alignment (Ji et al., 2025); surveys of direct preference optimization organize DPO-style methods by datasets, theoretical assumptions, variants, and applications (Xiao et al., 2024b; Liu et al., 2025e); and recent RL-for-LLM surveys provide broad coverage of RL methods across the LLM lifecycle or focus on the path toward reasoning models (Srivastava & Aggarwal, 2025; Liu et al., 2025c; Xu et al., 2025).

Complementary to these perspectives, our survey starts from the RL formulation of autoregressive language generation and traces its consequences for problem modeling, optimization algorithms, and training systems. This formulation highlights LLM-specific features—deterministic token-level transitions, sparse sequence-level feedback, large discrete action spaces, and rollout pipelines whose throughput and staleness affect feasible updates. We organize methods by how they respond to these features: how reward models, preference data, or verifiers define the optimization signal; how policy-gradient and preference-optimization methods balance variance reduction, stability, and computation; and how online/offline updates and distributed rollout systems shape sample efficiency and scalability. The resulting taxonomy is centered on algorithmic design and algorithm-system co-design trade-offs, rather than on application domains or method catalogs alone. We believe this formulation-first perspective can also help organize emerging questions on cost-effective RL, RL scaling laws, training dynamics, and the co-design of algorithms, infrastructure, and applications.

## 2 LLM and Its RL Formulation

This section reviews the background of Large Language Models (LLMs) and Reinforcement Learning (RL), establishing the mathematical framework that connects them. For clarity and ease of reference, a summary of important notations is provided in Table 1.

### 2.1 Background on LLMs

A language model is a generative model that operates by predicting the next token in a sequence through probabilistic modeling. At its core, an LLM employs a transformer architecture (Vaswani et al., 2017) to capture complex patterns and dependencies in textual data. Formally, an LLM is denoted as $\pi_\theta$, where $\theta \in \mathbb{R}^d$ represents the model's trainable parameters. The model selects tokens from a finite vocabulary $\mathcal{V} = \{1, \ldots, |\mathcal{V}|\}$, and given a context sequence $(a_1, \ldots, a_t)$, it models the conditional distribution of the next token as $a_{t+1} \sim \pi_\theta(\cdot|a_1, \ldots, a_t)$. This autoregressive generation continues until either a designated End-Of-Sentence (EOS) token appears or a predetermined maximum length $T$ is reached.

Modern LLMs have undergone substantial scaling, with parameter counts ranging from several billion (e.g., Llama-3-8B (Grattafiori et al., 2024) and Qwen-2.5-7B (Yang et al., 2024a)) to hundreds of billions (e.g.,

Table 1: Summary of Important Notations

| Notation | Description |
|---|---|
| *General Language Model Notations* | |
| $\pi_\theta$ | A Large Language Model (LLM) parameterized by $\theta$. |
| $\theta$ | The trainable parameters of the LLM. |
| $\mathcal{V}$ | The finite vocabulary of tokens. |
| $T$ | The maximum sequence length or horizon. |
| $x$ | The initial prompt or instruction sequence given to the model. |
| $y$ (or $y_{1:T}$) | A complete response sequence generated by the model. Used interchangeably with $a_{1:T}$. |
| | |
| *Reinforcement Learning (RL) Formulation Notations* | |
| $\mathcal{M}$ | A Markov Decision Process (MDP), defined as $(\mathcal{S}, \mathcal{A}, P, r, \rho, T)$. |
| $\mathcal{S}$ | The state space. In the LLM context, a state $s_t$ is a sequence of tokens. |
| $\mathcal{A}$ | The action space. In the LLM context, the vocabulary $\mathcal{V}$. |
| $s_t$ | The state at timestep $t$, typically the sequence $(x, a_1, \ldots, a_{t-1})$. |
| $a_t$ | The action at timestep $t$, corresponding to the token selected from $\mathcal{V}$. |
| $P$ | The state transition function. Deterministic in the LLM context. |
| $r$ | The reward function, assigning a scalar value to a state-action pair or a full sequence. |
| $\rho$ | The initial state distribution, representing the distribution of prompts $x$. |
| $\pi_{\text{ref}}$ | A reference policy, typically the SFT model, used for KL regularization. |
| $\beta$ | The regularization coefficient for the KL-divergence term. |

GPT-4 (OpenAI, 2023) and PaLM (Chowdhery et al., 2023)). This dramatic increase in model size has been empirically shown to correlate with emergent capabilities—advanced functionalities that are not observed in smaller-scale models (Wei et al., 2022a; Srivastava et al., 2022). While scaling has yielded impressive advancements in model performance, it simultaneously introduces significant challenges in computational efficiency, optimization strategies, and resource requirements, which we will examine in greater detail in subsequent sections. Before delving into the core of this survey, we provide a concise overview of the LLM training pipeline, as illustrated in Figure 2.

**Pre-training.** Unlike traditional deep learning models (e.g., AlexNet (Krizhevsky et al., 2012) and ResNet (He et al., 2016)) that are typically trained from scratch for specific tasks, LLMs are *pre-trained* on extensive corpora, optimizing for next-token prediction (Radford et al., 2019; Brown et al., 2020; Raffel et al., 2020). This process exposes models to diverse internet-scale datasets comprising books, articles, code repositories, and web pages—often totaling trillions of tokens (Gao et al., 2020; Penedo et al., 2023). The self-supervised nature of next-token prediction enables models to capture complex patterns, linguistic structures, and factual knowledge without requiring explicit labels, making this approach both scalable and effective for learning rich representations from raw text.

From another view, pre-training inherently implements a form of multi-task learning (Caruana, 1997; Radford et al., 2019), whereby a single model learns to perform multiple related tasks simultaneously. In the context of LLMs, these "tasks" correspond to different domains of knowledge, reasoning patterns, and linguistic capabilities (Wei et al., 2021; Sanh et al., 2021). To minimize the prediction loss, the model must learn to predict tokens in contexts as varied as mathematical proofs, fictional narratives, technical documentation, and conversational dialogues. This multi-domain training enables the development of generalized capabilities that transfer across tasks (Bommasani et al., 2021; Rae et al., 2021).

**Post-training.** Despite impressive capabilities in language understanding, pre-trained LLMs often struggle to produce contextually appropriate responses when applied directly to downstream applications, frequently generating repetitive or irrelevant content. This stems from a fundamental domain shift: while pre-training focuses on broad language modeling and understanding, downstream tasks require models to organize their knowledge and linguistic skills to generate well-structured, goal-oriented responses. To address these

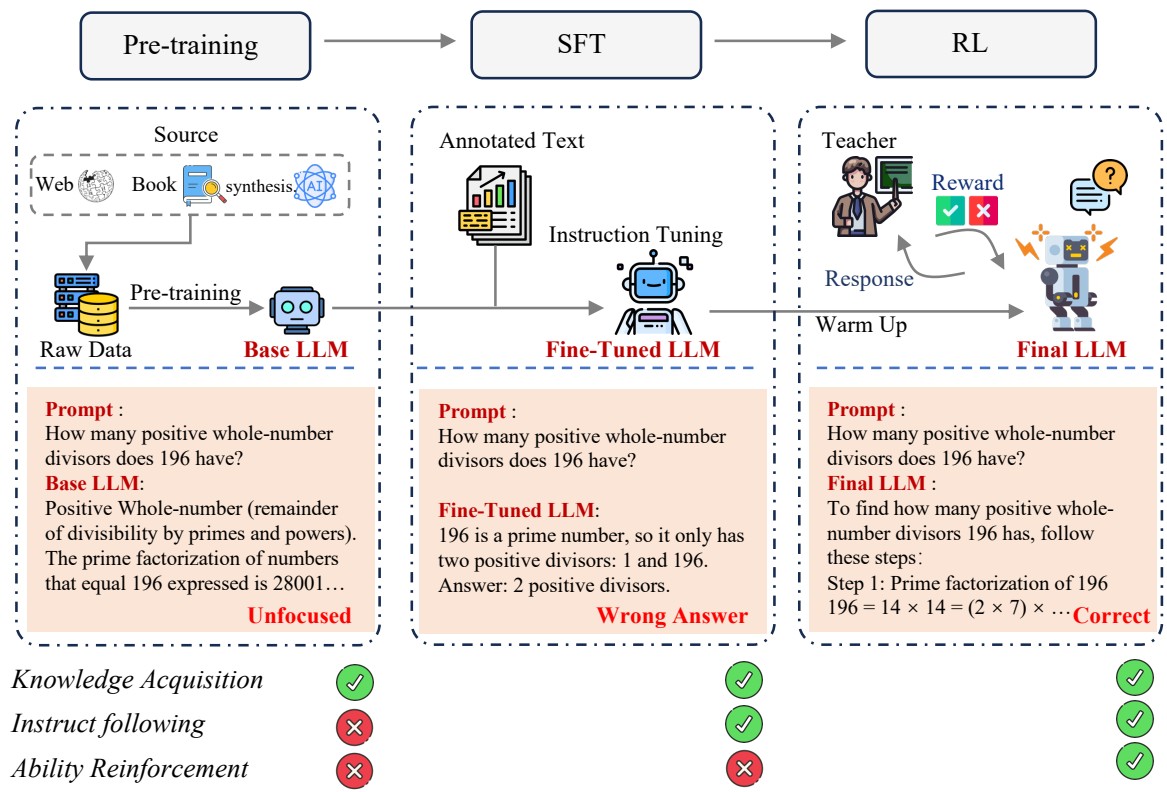

Figure 2: The three training stages of Large Language Models (LLMs): 1) Pre-training: A Base LLM is trained on vast raw data for next-token prediction, acquiring Knowledge Acquisition. Responses are typically unfocused and lack Instruction Following. 2) Supervised Fine-Tuning (SFT): The Base LLM is fine-tuned with instruction-response pairs, improving Instruction Following and structured output, though still prone to wrong answers in complex tasks. 3) Reinforcement Learning (RL): Optimized with human or AI feedback (reward signals), this stage dramatically enhances Ability Reinforcement, leading to a Final LLM that provides correct, well-reasoned, and aligned responses.

limitations, post-training methodologies have emerged as a crucial final development stage. As articulated by Zoph & Schulman (2025), post-training aims to "make the model behave like an assistant and follow the right format".

Post-training encompasses two primary directions of advancement. The first focuses on alignment with human values and preferences through Reinforcement Learning from Human Feedback (RLHF) (Christiano et al., 2017; Ziegler et al., 2019). This approach enables models to learn directly from human feedback, creating an iterative refinement process that better aligns responses with human expectations (Stiennon et al., 2020; Ouyang et al., 2022). The community now widely recognizes RLHF as a critical bridge between foundational pre-training capabilities and the practical requirements of reliable, helpful AI assistants.

The second direction enhances reasoning capabilities through techniques such as Chain-of-Thought reasoning (Wei et al., 2022b) and innovations in test-time scaling (Brown et al., 2024; Snell et al., 2024; Muennighoff et al., 2025), which leverage additional computational resources during inference to improve performance without further training. These advances have yielded remarkable results, exemplified by OpenAI's o-series model (OpenAI, 2024), which achieves 71.7% accuracy on the software engineering benchmark SWE-bench (Jimenez et al., 2023), 87.7% on graduate-level GPQA tasks (Rein et al., 2023), and 96.7% on competition-level mathematics reasoning (Hendrycks et al., 2021)—often surpassing human expert performance.

Crucially, RL techniques serve as the unifying mechanism across both directions, providing principled methods for optimizing model behavior and enhancing capabilities. The following section provides essential background on these RL principles and their applications to language model development.

## 2.2 Background on RL

In this section, we present the mathematical foundations of RL (Sutton & Barto, 2018) and subsequently demonstrate how this framework elegantly maps to the LLM optimization problem.

**Markov Decision Process.** RL is fundamentally grounded in the Markov Decision Process (MDP) formalism (Puterman, 2014), providing a principled approach to sequential decision-making under uncertainty. An MDP is formally represented as $\mathcal{M} = (\mathcal{S}, \mathcal{A}, P, r, \rho, T)$, where:

- $\mathcal{S}$ and $\mathcal{A}$ denote the state and action spaces, respectively;
- $P : \mathcal{S} \times \mathcal{A} \times \mathcal{S} \to [0, 1]$ defines the transition dynamics, with $s' \sim P(\cdot|s, a)$ representing the probability of transitioning to state $s'$ given current state $s$ and action $a$;
- $r : \mathcal{S} \times \mathcal{A} \to \mathbb{R}$ assigns reward values to state-action pairs;
- $\rho : \mathcal{S} \to [0, 1]$ represents the initial state distribution;
- $T$ denotes the finite horizon or episode length.

The central objective in the MDP framework is to identify a policy $\pi_\theta$ that maximizes the expected cumulative reward over the specified horizon $T$:

$$\max_\theta \mathbb{E}_{s_1 \sim \rho} \mathbb{E}_{a_{1:T} \sim \pi_\theta} \left[ \sum_{t=1}^{T} r(s_t, a_t) \mid s_{t+1} \sim P(\cdot|s_t, a_t) \right]. \tag{1}$$

**Specification in LLMs.** When applying this framework to LLMs, we establish a direct correspondence between MDP components and LLM generation processes. Following (Li et al., 2024c; Rafailov et al., 2024b), a state $s_t$ encompasses the sequence of previously generated tokens, represented as $s_t = (x, a_1, \ldots, a_{t-1})$, where $x$ denotes the initial prompt. An action $a_t$ corresponds to selecting a token from the vocabulary set $\mathcal{V}$. The initial state $s_1$ is simply the prompt $x$, with $\rho$ reflecting the prompt distribution. We note that in this formulation, the action space is large (e.g., $128K$ for Llama-3), and recent work explores more compact action spaces (Jia et al., 2025; Kim et al., 2025b).

A crucial distinction between LLM-based MDPs and traditional RL problems lies in the nature of state transitions. Unlike conventional RL settings where transitions are typically stochastic due to environmental uncertainty, the MDP formulation for LLMs features **deterministic transitions**—a property leveraged by Li et al. (2024c) to substantially simplify the RL optimization procedure. Specifically, the deterministic transition function $P$ merely appends the current token to the history:

$$P(s_{t+1}|s_t, a_t) = \begin{cases} 1 & \text{if } s_{t+1} = (s_t, a_t) \\ 0 & \text{otherwise.} \end{cases} \tag{2}$$

Here, $(s_t, a_t)$ denotes the concatenation of the state $s_t$ with action $a_t$, effectively creating the next state by extending the token sequence.

This determinism contrasts sharply with traditional RL challenges (Sutton & Barto, 2018), where executing identical actions from the same state can yield different outcomes due to environmental stochasticity. Stochastic transition RL problems are inherently difficult to solve (Azar et al., 2017), as this stochasticity lies beyond the learner's control, typically requiring advanced techniques such as value networks for variance reduction (Cai et al., 2020). The deterministic nature of LLM state transitions eliminates this significant source of complexity, enabling more efficient algorithms specifically tailored to language model fine-tuning, as we will review in subsequent sections.

In the context of LLMs, we evaluate complete responses holistically rather than assigning rewards to individual tokens. This approach requires adapting the traditional MDP framework by concentrating the entire reward signal at sequence completion while assigning zero rewards to all intermediate generation steps:

$$r(s_t, a_t) = \begin{cases} 0 & \text{if } t \neq T \\ r(x, a_{1:T}) & \text{otherwise.} \end{cases} \tag{3}$$

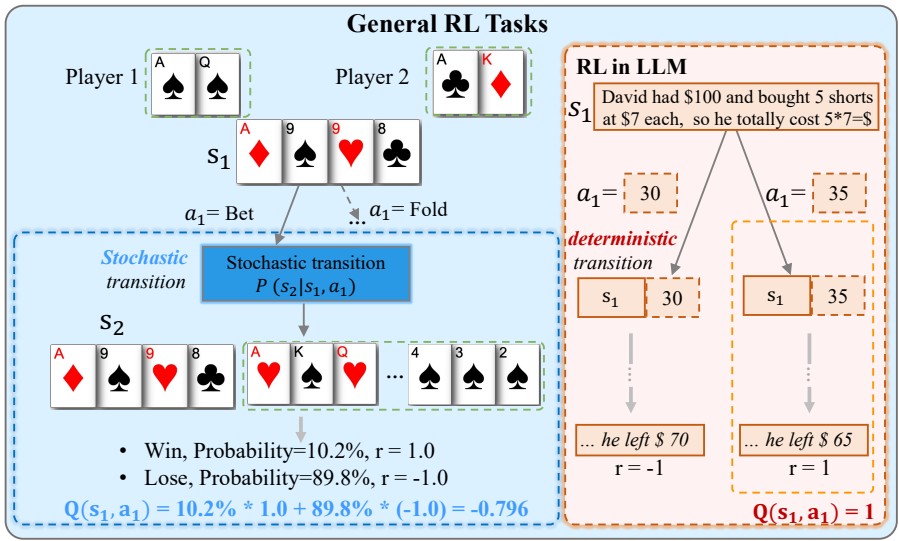

Figure 3: The fundamental distinction between general RL tasks and RL in LLMs. On the left, in a traditional RL poker setting, the state $S_1$ represents the current game situation, and an action $a_1$ (e.g., Bet) leads to a stochastic transition to a new state $S_2$, reflecting environmental uncertainty. The $Q$-value, $Q(s_1, a_1)$, calculates the expected future return (win rate) considering these probabilistic outcomes. On the right, in the LLM setting, the initial state $S_1$ is the prompt ($x$), and an action $a_1$ corresponds to selecting a token from the vocabulary. A crucial characteristic here is the deterministic transition: appending a chosen token to the current state always results in a unique, predictable next state. Additionally, LLMs typically employ a sparse reward function, where rewards (e.g., $r = 1$ or $r = -1$) are assigned only at the completion of a full response sequence, with intermediate token generation steps receiving zero reward.

For instance, this sparse reward function might evaluate the helpfulness and harmlessness of an entire response in alignment tasks, or assess final answer accuracy in mathematical reasoning problems.

Under these specifications, the core optimization objective for LLMs becomes a straightforward reward maximization problem:

$$\max_\theta \mathbb{E}_{x \sim \rho} \mathbb{E}_{a_{1:T} \sim \pi_\theta} [r(x, a_{1:T})], \tag{4}$$

That is, we seek parameters $\theta$ that maximize the expected reward across the distribution of prompts and model-generated responses. When the context is clear, we simply exchange the notation of $a_{1:T}$ and the response $y_{1:T}$ for conciseness.

Note that our current formulation, particularly the definition of a single reward at the end of a fixed-length sequence ($T$), implicitly focuses on single-turn response generation. This is a common and foundational setting for many tasks, where the model receives an initial prompt ($x$) and generates one complete, self-contained response ($a_{1:T}$). We notice that the real-world deployment of LLMs frequently involves more complex, sequential interactions. In multi-turn conversational settings (see e.g., (Shani et al., 2024; Xiong et al., 2024)) or sophisticated agentic tasks (see e.g., (Singh et al., 2025; Dong et al., 2025; Xue et al., 2025)), the MDP framework requires extensions. Specifically, the state space ($\mathcal{S}$) would expand significantly. Instead of just the current prompt and partial response, $s_t$ would need to encompass the entire conversational history up to that point. This introduces challenges related to context window management and long-term memory (Yu et al., 2025a). Furthermore, the concept of an action ($a_t$) becomes multi-layered. While selecting an individual token from $\mathcal{V}$ remains the fundamental micro-action, the generation of an entire response for a turn could be considered a "macro action" within a higher-level decision process. The reward function ($r$) may typically shift from a single, end-of-sequence signal to potentially being received after each turn, or even more sparsely after multiple turns, depending on the dialogue's objective. This introduces the significant challenge of credit assignment over extended interaction sequences (Zeng et al., 2025a). Nevertheless, these

extended scenarios often build upon the fundamental techniques established in single-turn settings, which remains the primary focus of this survey.

## 2.3 Formulations

In this section, we introduce three key formulations and tasks in post-training with SFT and RL, and connect them with the terminology defined before.

**SFT.** As its name suggests, SFT often involves fine-tuning a model with supervised data using the following likelihood-maximization objective:

$$\max_{\theta} \sum_{i=1}^{n} \log \pi_{\theta}(y_{1:T}^{i}|x^{i}) \tag{5}$$

where $x^i$ represents the prompt and instruction, $y_{1:T}^i$ denotes the corresponding completion to be learned, and $n$ is the number of training samples. Note that while $x^i$ is also a sequence of tokens, we omit the timestep subscript for notational clarity.

What is the functional role of SFT, and what practical insights does the likelihood-maximization objective provide? Answering this question requires examining why pre-trained LLMs struggle with downstream applications. Pre-trained models often exhibit problematic behaviors, including hallucinating facts, generating biased or toxic content, and failing to follow user instructions (Bender et al., 2021; Weidinger et al., 2021). This misalignment stems from a fundamental objective mismatch (Ouyang et al., 2022): pre-training optimizes next-token prediction on internet text, which differs substantially from the desired goal of "following user instructions helpfully and safely". SFT addresses this misalignment by maintaining the same technical objective (next-token prediction) while fundamentally shifting the training distribution from general text corpora to curated instruction-response pairs and task-specific datasets (Raffel et al., 2020; Wei et al., 2021). This data shift represents a crucial step toward alignment, transforming the model's learned distribution to better match human expectations.

From an RL perspective, SFT serves a critical bridging function: it transforms model outputs into a form that human evaluators or reward models can effectively understand and assess, thereby enabling meaningful reward signals during subsequent RL training (Li et al., 2025g; Zeng et al., 2025b). While direct RL on pre-trained models without SFT is technically possible, as demonstrated by Guo et al. (2025), such approaches often result in models that develop problem-solving capabilities while producing confusing or incoherent outputs that are difficult for humans to interpret.

Why does SFT deserve recognition as a distinct approach? Beyond being "just supervised learning," SFT operates in an unusual post-pretraining regime where (i) a small amount of curated instruction data can strongly steer formatting and interaction style, and (ii) the objective (token-level likelihood) can inadvertently compress output diversity. Concretely, cross-entropy fine-tuning may reduce response entropy and suppress minority styles or non-mainstream continuations, motivating implicit entropy regularization and other diversity-preserving techniques (Li et al., 2025g).

**RLHF.** The framework of RLHF in LLMs was proposed in (Stiennon et al., 2020; Ouyang et al., 2022). The goal of RLHF is to further align the model with human preferences based on a preference dataset $\mathcal{D}^{\mathrm{pref}}$. The preference dataset has the form $\mathcal{D}^{\mathrm{pref}} = \{(x, y_{1:T}^w, y_{1:T}^l)\}$, where the response $y_{1:T}^w$ is preferred over $y_{1:T}^l$. The RLHF framework typically involves two steps: reward learning and policy learning. To infer the reward, most existing works (Bai et al., 2022a; Ouyang et al., 2022; Rafailov et al., 2023) leverage the following Bradley-Terry (BT) model assumption (Bradley & Terry, 1952).

**Assumption 1** (Bradley-Terry (BT) Model)**.** *The Bradley-Terry model defines the preference distribution as*
$$\mathbb{P}(y_{1:T} \succ y'_{1:T}|x) = \sigma(r^\star(x, y_{1:T}) - r^\star(x, y'_{1:T})),$$
*where $\sigma(\cdot)$ denotes the sigmoid function and $r^\star$ represents the unknown true reward.*

Under the BT model assumption, one can explicitly (Bai et al., 2022a; Ouyang et al., 2022) or implicitly (Rafailov et al., 2023) learn the reward by applying maximum likelihood estimation on the preference dataset.

$$\widehat{r} = \underset{r}{\arg\max} \, \mathbb{E}_{(x, y_{1:T}^w, y_{1:T}^l) \sim \mathcal{D}^{\mathrm{pref}}} \left[ \log \left( \sigma(r(x, y_{1:T}^w) - r(x, y_{1:T}^l)) \right) \right].$$

This equation presents the most basic formulation for reward learning; for the latest advancements, please refer to (Lambert et al., 2024; Wang et al., 2024a; Malik et al., 2025). Beyond this discriminative reward learning paradigm, an emerging research direction has focused on generative reward models (Zhang et al., 2024a; Mahan et al., 2024; Liu et al., 2025h). These models leverage chain-of-thought (Wei et al., 2022b) for reward judgment, offering an alternative approach to traditional discriminative methods.

When training a policy with reward models, flaws in the reward model can lead to reward hacking (Gao et al., 2022; Guo et al., 2025; Rafailov et al., 2024a; Chen et al., 2024), where the policy exploits weaknesses in the reward model to achieve high estimated rewards without genuine performance improvements. In particular, due to the limited size and coverage of the preference data, the learned reward model may deviate from the true reward (Zhu et al., 2023), especially in regions outside the preference data distribution. This phenomenon arises from reward misgeneralization, where reward models compute rewards using spurious features irrelevant to human preferences (Miao et al., 2024). The consequences of reward hacking pose significant safety concerns for LLMs, particularly in high-stakes applications. Research has shown that models can learn to manipulate human feedback through RLHF, generating responses that appear correct and convincing to humans while being factually incorrect (Wen et al., 2024). Beyond performance degradation, reward hacking enables models to exploit limitations in human attention or knowledge, potentially leading to the propagation of misinformation in critical domains. These vulnerabilities are further exacerbated by adversarial attacks, where carefully crafted jailbreak prompts can circumvent safety measures even in well-aligned models (Zou et al., 2023; Ganguli et al., 2022).

**Whose preferences and what biases are optimized?** While RLHF is often described as alignment to "human preferences", the learned objective is necessarily *population- and protocol-dependent.* Preferences in $\mathcal{D}^{\mathrm{pref}}$ reflect who is recruited, what instructions raters receive, and how disagreements are aggregated; consequently, RLHF can privilege majority norms, culturally-specific notions of "helpful", or institutional risk tolerances, raising the "whose values?" problem (Barnhart et al., 2025; Dahlgren Lindström et al., 2025). In addition, both reward models and the benchmarks used to evaluate them exhibit systematic biases (e.g., style/verbosity and length effects), and *optimization tends to amplify whatever the reward model measures well*—including spurious correlates—because policy learning concentrates probability mass on high-scoring regions (Gao et al., 2022; Lambert et al., 2024; Mac Kim et al., 2025). Practically, this motivates (i) reporting annotator and preference-collection details (Ouyang et al., 2022; Bai et al., 2022a), (ii) stress-testing reward models under distribution shift (Coste et al., 2023; Eisenstein et al., 2023), and (iii) conservative optimization (e.g., KL control, ensembles, and red-teaming) to reduce bias amplification during training (Stiennon et al., 2020; Ganguli et al., 2022; Mazeika et al., 2024; Zhang et al., 2024b).

Among these strategies, KL regularization has become the most widely adopted approach for conservative optimization. By penalizing deviation from a reference policy, it prevents the model from over-exploiting potentially spurious reward signals:

$$\max_{\theta} \mathbb{E}_{x \sim \rho} \left[ \mathbb{E}_{a_{1:T} \sim \pi_\theta(\cdot|x)}[\widehat{r}(x, a_{1:T})] - \beta D_{\mathrm{KL}}(\pi_\theta(\cdot|x), \pi_{\mathrm{ref}}(\cdot|x)) \right].$$

Here $\beta > 0$ is the regularization coefficient and $\pi_{\mathrm{ref}}$ is a reference policy, typically chosen as the policy model after SFT. Beyond KL regularization, recent research has explored several complementary mitigation strategies. Reward model ensembles combine multiple reward models with conservative optimization objectives (Coste et al., 2023; Eisenstein et al., 2023). Information-theoretic approaches filter spurious features via variational information bottlenecks (Miao et al., 2024). Constrained optimization frameworks explicitly separate helpfulness and safety objectives using separate reward and cost models (Dai et al., 2023). Additionally, out-of-distribution detection mechanisms (Bukharin et al., 2025) and systematic red-teaming (Ganguli et al., 2022; Zou et al., 2023; Wang et al., 2025b) have become essential for identifying vulnerabilities before deployment. These complementary approaches collectively enhance the safety and reliability of RLHF systems; for a comprehensive survey, please refer to (Ji et al., 2025).

**Reinforcement Learning with Verifiable Reward.** A driving force behind recent breakthroughs in LLM reasoning—including OpenAI-o1 (OpenAI, 2024), DeepSeek-R1 (Guo et al., 2025), and Kimi-1.5 (Kimi et al., 2025)—is Reinforcement Learning with Verifiable Reward (RLVR). Unlike RLHF, which relies on learned reward models, RLVR trains LLMs by applying RL to maximize a *rule-based outcome reward*:

$$\max_\theta \mathbb{E}_{x \sim \rho} \left[ \mathbb{E}_{a_{1:T} \sim \pi_\theta(\cdot|x)}[r_{\text{rule}}(x, a_{1:T})] \right] - \beta D_{\text{KL}}(\pi_\theta(\cdot|x), \pi_{\text{ref}}(\cdot|x)).$$

This rule-based reward $r_{\text{rule}}$ evaluates the correctness of the final answer based on deterministic verification procedures. For mathematical tasks, one can directly compare the final answer in the response with the ground-truth answer, checking for mathematical equivalence. For coding tasks, one can leverage a compiler or interpreter to evaluate whether the generated code successfully passes a suite of test cases, providing binary feedback on functional correctness.

**Comparison with RLHF.** Because the reward is grounded in an explicit verifier rather than a learned model, RLVR can substantially reduce classic reward hacking driven by reward misgeneralization in unobserved regions. However, RLVR remains vulnerable to *specification gaming*: the verifier and its "golden rules" encode designer choices about what counts as success (e.g., pass@1 vs. pass@k, formatting constraints, unit-test coverage, or mathematical equivalence conventions), which can privilege certain solution styles and failure modes. Consequently, realizing the full potential of RLVR requires careful verifier design, adversarial test generation, and complementary evaluations beyond the training-time verifier to ensure generalization. A natural question that arises is whether providing feedback on intermediate reasoning steps, rather than only the final outcome, could further improve learning efficiency and error localization.

**Process Reward Models: A Comparative Perspective.** Process Reward Models (PRMs) address this question by providing feedback for each intermediate reasoning step rather than only the final answer (Lightman et al., 2023a; Uesato et al., 2022). This denser supervision can precisely localize errors within a reasoning chain, making PRMs particularly appealing for complex multi-step problems. Notable examples include Math-Shepherd (Wang et al., 2024b), which automatically constructs process-wise supervision using Monte Carlo estimation to assess whether each step can lead to a correct final answer, and PRM800K (Lightman et al., 2023a), a large-scale dataset of 800,000 human-annotated step-level labels. Empirically, PRMs have demonstrated strong performance in verification and reranking settings, with Lightman et al. (2023a) showing that process supervision significantly outperforms outcome supervision on the challenging MATH dataset.

Despite these promising results, PRMs face notable scalability challenges that currently limit their practical deployment in large-scale RL training (Guo et al., 2025). First, defining fine-grained reasoning steps is inherently ambiguous for general reasoning tasks. Second, determining step-level correctness is difficult: automated annotation methods often yield noisy labels, while manual annotation is prohibitively expensive and does not scale. Third, introducing learned PRMs risks the same reward hacking vulnerabilities as standard RLHF and requires periodic retraining, substantially increasing computational costs. Recent work has explored mitigating these challenges through implicit PRMs that derive step-level rewards from outcome-level training (Yuan et al., 2024), though the practical advantages of explicit process supervision remain limited at scale. Given these trade-offs, current state-of-the-art reasoning models such as DeepSeek-R1 rely primarily on outcome-based verifiable rewards, which offer simpler and more scalable supervision for large-scale RL (Guo et al., 2025). Nevertheless, PRMs remain an active research frontier, with ongoing efforts to reduce annotation costs and improve robustness potentially enabling their broader adoption in future systems.

## 2.4 Challenges of RL in LLMs

In this section, we highlight several unique challenges in applying RL to train LLMs, which calls for the development of more effective and efficient RL algorithms tailored for LLMs. We highlight two notable challenges below.

**Challenge: Huge Model Size.** The first major challenge stems from the unprecedented scale of modern LLMs, driven by influential scaling laws (Kaplan et al., 2020) that demonstrate consistent performance improvements with increased model parameters across diverse tasks. Following this principle, LLMs have

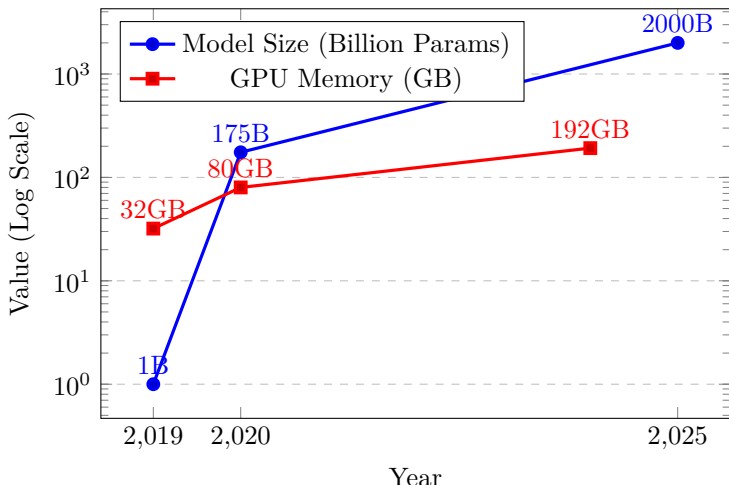

Figure 4: A comparison of LLM model size and single-GPU memory growth, plotted on a logarithmic scale. Model parameters have grown exponentially, from 1.5B (GPT-2, 2019) to 175B (GPT-3, 2020) and a projected 2,000B (Llama-4 Behemoth, 2025). In stark contrast, the memory of flagship GPUs has increased at a much slower, near-linear rate, from 32GB (V100, 2019) to 80GB (A100, 2020) and 384GB (GB200, 2024). This widening gap illustrates a critical hardware bottleneck that complicates the memory-intensive process of RL training for large-scale models.

undergone dramatic expansion: GPT-2 contained 1 billion parameters in 2019 (Radford et al., 2019), GPT-3 (Brown et al., 2020) scaled to 175 billion parameters in 2020, and recent models like Llama-4 Behemoth have reached 2 trillion parameters by 2025 (Meta, 2025). This exponential growth in model size creates unique computational demands that distinguish LLM RL training from traditional RL applications.

RL training significantly amplifies these computational challenges compared to standard supervised learning. Unlike supervised training that requires only a single model instance, RL algorithms such as PPO (Schulman et al., 2017) typically maintain multiple model copies simultaneously—including actor networks, critic networks, and reference models for KL regularization—effectively multiplying memory requirements by approximately 2-3 times. For trillion-parameter models, this expanded memory footprint can exceed the capacity of even the most advanced GPU clusters. Moreover, RL algorithms require multiple forward and backward passes per training step, with PPO performing 4-8 gradient steps per batch of collected samples, substantially increasing computational overhead compared to single-pass supervised training.

This challenge is exacerbated by a growing divergence between model scaling and hardware capabilities. As illustrated in Figure 4, model parameter counts have grown exponentially while the memory of individual GPUs has scaled at a much slower, near-linear pace. For instance, flagship NVIDIA GPU memory increased 6-fold from 32GB (V100) to 192GB (B200) between 2019 and 2024. Over a comparable timeframe, state-of-the-art model sizes increased by more than 1,000-fold (from 1.5B to a projected 2,000B parameters). This widening gap underscores the urgent need for memory-efficient RL algorithms (Rafailov et al., 2023; Li et al., 2024c) and novel distributed training infrastructures to make large-scale RL optimization feasible.

**Energy and sustainability.** Large-scale RL also has a nontrivial energy footprint. Beyond standard backpropagation costs, online RL repeatedly performs high-throughput sampling (often with multiple rollouts per prompt) and may maintain additional model copies (e.g., reference policies and auxiliary components), increasing total compute relative to SFT for the same model size. At the scale of frontier models, the resulting electricity use and associated emissions become a meaningful systems constraint and a broader societal concern (Patterson et al., 2021). This motivates reporting compute and energy proxies for RL runs when possible, and prioritizing algorithmic/system advances that reduce sampling cost (e.g., better credit assignment, more sample-efficient objectives, and faster inference backends).

**Challenge: Data Heterogeneity.** Beyond computational constraints, the heterogeneous nature of training data presents significant challenges. LLMs are typically trained on diverse datasets encompassing multiple

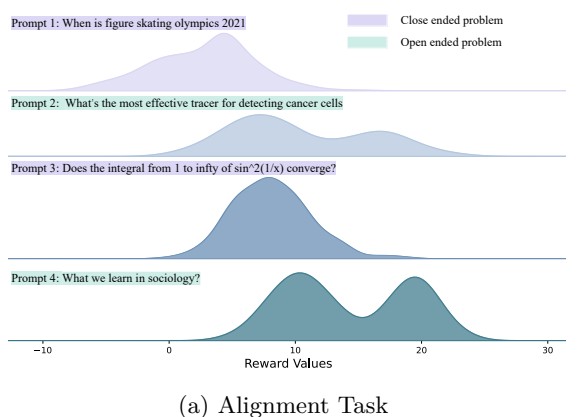
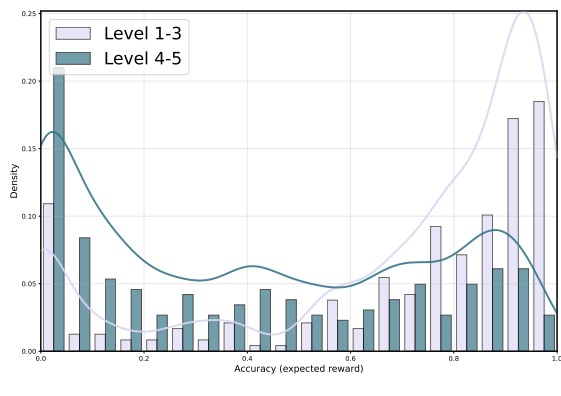

(a) Alignment Task

(b) Math Reasoning Task

Figure 5: Heterogeneity in training data distribution across task types. (a) Alignment tasks show four distinct reward distributions corresponding to different prompt categories (open-ended problem vs. close-ended problem), with reward values ranging from approximately -10 to 30. (b) Mathematical reasoning tasks exhibit high variability in expected rewards (accuracy) across different prompts, with accuracy ranging from 0.0 to 1.0. The bimodal distribution in panel (b) suggests two distinct performance regimes, with Level 1-3 tasks (light bars) generally achieving higher accuracy than Level 4-5 tasks (dark bars). The overlaid density curves illustrate the substantial heterogeneity in performance expectations across the prompt distribution for both task categories.

domains and task types, each exhibiting substantially different reward characteristics (Ouyang et al., 2022). This represents a one-model-for-many-tasks paradigm, contrasting sharply with the one-model-for-one-task approach common in classical RL applications (Mnih et al., 2015; Lillicrap et al., 2015).

This heterogeneity manifests across multiple dimensions of task complexity and domain specificity. In preference alignment tasks, reward distributions vary dramatically between prompt categories: open-ended creative prompts typically yield moderate, multi-modal distributions, while factual, closed-ended prompts such as "What is the capital of New Zealand?" often produce more extreme reward values due to their binary correct/incorrect nature (Song et al., 2023). Similarly, in RLVR for reasoning tasks, different mathematical problem difficulty levels exhibit vastly different success rates and corresponding reward distributions. The empirical evidence of this heterogeneity is illustrated in Figure 5, which shows distinct reward distribution patterns across alignment tasks (panel a) and mathematical reasoning tasks of varying difficulty levels (panel b). In alignment tasks, we observe four distinct reward distributions corresponding to different prompt categories, with reward values ranging from approximately -10 to 30. For mathematical reasoning tasks, the bimodal distribution reveals two distinct performance regimes: Level 1-3 tasks (light bars) generally achieve higher accuracy than Level 4-5 tasks (dark bars), with accuracy ranging from 0.0 to 1.0. These substantial distributional differences across task types create fundamental challenges for unified RL optimization.

This reward distribution heterogeneity poses critical challenges for RL training dynamics and model convergence (Li et al., 2024c). The varying scales and distributions can lead to gradient imbalance effects, where tasks with higher reward variance dominate gradient updates, potentially causing catastrophic forgetting in other domains or suboptimal allocation of learning capacity. Tasks with extreme rewards may receive disproportionate attention during optimization, while those with moderate but important improvements may be neglected. Furthermore, conflicting gradient signals from different task types can cause oscillatory training behavior or prevent proper convergence, necessitating specialized RL algorithms designed to handle multi-task gradient rebalancing and ensure stable optimization across heterogeneous reward landscapes.

The two challenges mentioned above directly hinder the scaling of RL for large models and datasets. Beyond these primary issues, RL training may also face several additional complexities: sparse, delayed reward structures that complicate credit assignment (Lightman et al., 2023b; Cui et al., 2025a); difficult exploration and exploitation in extremely high-dimensional discrete action spaces (vocabularies of 50,000+ tokens) (Ramamurthy et al., 2022; Jia et al., 2025); and difficulties in accurate value function approximation over high-dimensional discrete text sequence spaces (Yuan et al., 2025b). While we cannot provide a comprehensive

Table 2: Compact comparison of the main optimization classes for LLM post-training. RLHF and RLVR are treated as feedback/problem formulations rather than algorithm families. The Value model and Online rollout columns use tick/cross indicators.

| Alg. class | Algorithms | Feedback type | Training regime | Value model | Online rollout |
|---|---|---|---|---|---|
| Actor–critic PG | PPO, TRPO, A3C | Reward model or rule-based outcome reward | On-policy | ✓ | ✓ |
| Critic-free PG | REINFORCE, ReMax, GRPO, RLOO | Reward model or rule-based outcome reward | On-policy | ✗ | ✓ |
| Preference Opt. | DPO, IPO, KTO | Preferences | Offline | ✗ | ✗ |

discussion of each individual challenge, we note that some are inherited from classical RL and established wisdom may inspire viable solutions.

## 3 RL Algorithms

In this section, we review leading RL optimization methods (Schulman et al., 2017; Li et al., 2024c; Ahmadian et al., 2024; Shao et al., 2024; Rafailov et al., 2023) for LLM post-training. We organize these methods into two categories aligned with the RL formulation used in this survey: *online policy-gradient* methods, which iteratively collect model-generated rollouts, and *preference-based offline optimization* methods, which update the policy from fixed preference data without online rollout infrastructure.

Before presenting individual algorithms, Table 2 gives a compact view of the optimization classes most relevant to LLM post-training. We separate feedback/problem formulations from optimization mechanisms: RLHF and RLVR define learning signals such as learned reward models or rule-based/verifier rewards (Ouyang et al., 2022; Shao et al., 2024), while actor–critic policy gradients, critic-free policy gradients, and preference optimization define how the policy is updated from that signal or data. The table focuses on four design-relevant dimensions—feedback type, training regime, value-model use, and online rollout use—which shape the stability and computational cost discussed below.

The main trade-offs depend on how a feedback formulation is paired with an optimization mechanism. Actor–critic methods such as TRPO, PPO, and A3C (Schulman et al., 2015; 2017; Mnih et al., 2016) use a value model to reduce variance and stabilize policy-gradient updates, but add memory, synchronization, and wall-clock overhead. Critic-free methods, from REINFORCE to recent LLM-oriented variants such as ReMax, GRPO, and RLOO (Williams, 1992; Li et al., 2024c; Shao et al., 2024; Ahmadian et al., 2024), remove this auxiliary model, improving efficiency while making baseline design more important. Preference-optimization methods such as DPO, IPO, and KTO (Rafailov et al., 2023; Azar et al., 2024; Ethayarajh et al., 2024) avoid online rollout infrastructure when preference data is available, but their reliability depends on data coverage and distribution shift. These distinctions motivate the following subsections on online policy-gradient and preference-based methods.

### 3.1 Online Algorithms

At a high level, online RL algorithms for LLMs involve two fundamental conceptual processes: exploration and exploitation (Sutton & Barto, 2018; Agarwal et al., 2019). In the context of LLMs, exploration is the process of sampling diverse responses from the model to discover high-value outcomes and gather informative training data. Exploitation, in turn, involves learning from the collected data to improve the model's policy. This iterative process is illustrated in Algorithm 1.
The exploration component in LLMs is conceptually straightforward: it typically involves sampling multiple responses for each prompt according to the current language model distribution $\pi_\theta$. This stochastic sampling naturally provides the diversity needed to explore the response space. However, the exploitation

---

**Algorithm 1** Online RL Framework for LLM Training

---

**Input:** Language model $\pi_\theta$, reward function/model $r : \mathcal{X} \times \mathcal{Y} \to \mathbb{R}$
  **for** training iteration $1, 2, \ldots$ **do**
    **Exploration:** Sample multiple responses for a mini-batch of prompts from $\pi_\theta$
    **Evaluation:** Calculate reward values $r(x, y)$ for each prompt-response pair
    **Exploitation:** Compute gradient estimator using collected data and rewards
    **Update:** Perform gradient ascent step to update policy parameters $\theta$

---

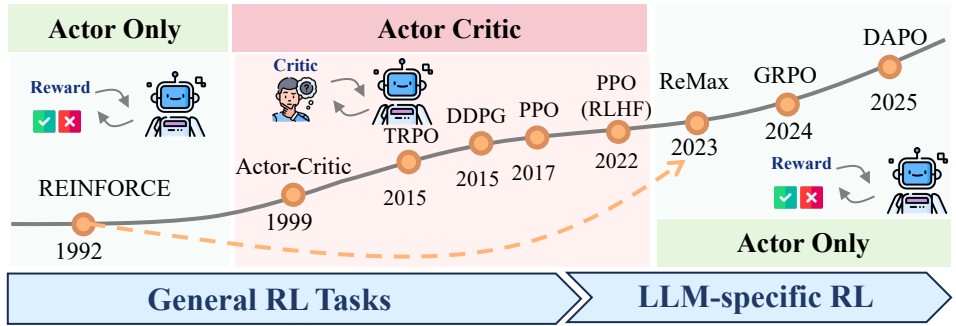

Figure 6: Evolution of RL algorithms. Early RL methods progressed from REINFORCE to actor-critic methods and then to PPO. With the rise of RL for LLMs, there has been renewed interest in REINFORCE-style methods, because they are (1) computationally cheaper and easier to scale, and (2) in the standard single-turn LLM formulation with deterministic token transitions, they avoid a major source of variance that motivates value-function-heavy methods in classical RL.

phase—effectively updating the model parameters using the collected data to maximize performance—presents significant challenges (Ramamurthy et al., 2022; Li et al., 2024c). In this section, we focus primarily on this crucial aspect, discussing key insights for designing gradient estimators that enable stable and effective policy updates. To start, we first review a classical result from the RL literature.

**Theorem 1** (Policy Gradient Theorem Sutton (1988)). *The exact gradient of reward maximization in Equation* (4) *can be computed as follows:*

$$\nabla_\theta \mathbb{E}_{x \sim \rho} \mathbb{E}_{a_{1:T} \sim \pi_\theta}[r(x, a_{1:T})] = \mathbb{E}_{x \sim \rho} \sum_{a_{1:T}}[r(x, a_{1:T}) \cdot \nabla_\theta \pi_\theta(a_{1:T}|x)] \tag{6}$$

$$= \mathbb{E}_{x \sim \rho} \mathbb{E}_{a_{1:T} \sim \pi_\theta}[r(x, a_{1:T}) \cdot \nabla_\theta \log \pi_\theta(a_{1:T}|x)]. \tag{7}$$

*Furthermore, a simple stochastic gradient estimator is*

$$\widetilde{g}(\theta) = \sum_{i=1}^{n} r(x^i, a_{1:T}^i) \cdot \nabla_\theta \log \pi_\theta(a_{1:T}^i|x^i) \quad \text{with } x^i \sim \rho, a_{1:T}^i \sim \pi_\theta(\cdot|x^i). \tag{8}$$

The key insight of the policy gradient theorem lies in the transition from Equation (6) to Equation (7), where the expectation shifts from a uniform distribution to the current model distribution $\pi_\theta$. This transformation enables the accurate stochastic estimation formulation presented in Equation (8).

At a high level, there are two main considerations in designing policy gradient estimators. One is *stability*, which primarily means seeking an estimator with low variance that enables reliable and stable performance improvements after gradient steps (Kakade, 2001; Schulman et al., 2015). Ensuring such stability is a minimal requirement for successful training, preventing issues like divergence during optimization. Following this foundational aspect, another important consideration is *regularization*, particularly *entropy regularization* (Haarnoja et al., 2018; Cui et al., 2025b). While entropy regularization can introduce a bias away from purely optimal policies, it incentivizes output diversity and enhances exploration. With these considerations in mind, we first review methods focused on improving stability by reducing variance. Subsequently, we discuss emerging studies that additionally incorporate entropy regularization.

**History Overview.** The first algorithm to use stochastic gradients for policy optimization is REINFORCE (Williams, 1992), introduced in the 1990s and widely referenced in the literature. Specifically, it builds on the stochastic gradient estimator in Equation (8). A key feature of REINFORCE is its simplicity of implementation, as it does not require a separate so-called value function. However, it has historically shown poor performance in classical deep RL applications (such as games and robotic control) due to the high variance of its stochastic gradient estimates (Sutton & Barto, 2018; Li et al., 2024c). Consequently, from the 2000s to the 2020s, the RL community developed actor-critic algorithms (Konda & Tsitsiklis, 1999; Lillicrap et al., 2015; Haarnoja et al., 2018) that provide lower-variance gradient estimates. The key idea behind these methods is that introducing a value model can accumulate learning from historical data, thereby reducing uncertainty and the variance of stochastic gradients. One such prominent algorithm is Proximal Policy Optimization (PPO) (Schulman et al., 2017). Recently, Li et al. (2024c) have argued that the variance of direct policy gradient estimation in REINFORCE can be significantly reduced when working with deterministic transitions—as in LLM fine-tuning, where external environment noise is absent—making REINFORCE viable again while retaining its computational simplicity. Figure 6 provides an overview of this algorithmic evolution, and in the following sections, we provide a detailed review of these methods.

**PPO.** Widely used in deep RL, PPO has proven successful in complex domains ranging from video games to robotics control. We will first review PPO's standard state-action formulation before adapting it to language generation tasks. Building upon natural policy gradient (Kakade, 2001) and trust-region-based policy optimization (Schulman et al., 2015), PPO optimizes a surrogate function:

$$\mathcal{L}_{\text{ppo}} = \mathbb{E}_{x\sim\rho}\mathbb{E}_{a_{1:T}\sim\pi_{\theta_{\text{old}}}}\left[\sum_{t=1}^{T}\widetilde{A}(s_t,a_t)\min\left\{\psi(s_t,a_t),\text{clip}\left(\psi(s_t,a_t),1-\delta,1+\delta\right)\right\}\right], \tag{9}$$

$$\psi(s_t,a_t) = \frac{\pi_\theta(a_t|s_t)}{\pi_{\theta_{\text{old}}}(a_t|s_t)}, \tag{10}$$

$$\widetilde{A}(s_t,a_t) = \sum_{j=0}^{T-t}\lambda^j\texttt{advantage}_{t+j} = \sum_{j=0}^{T-t}\lambda^j[r(s_{t+j},a_{t+j})+\gamma V(s_{t+1+j})-V(s_{t+j})]. \tag{11}$$

$$\mathcal{L}_{\text{value}} = \mathbb{E}_{x\sim\rho}\mathbb{E}_{a_{1:T}\sim\pi_{\theta_{\text{old}}}}\left[(V(s_t,a_t)-y_t)^2\right]$$

$$y_t = \text{stop\_gradient}\left\{\sum_{\ell=1}^{T-t}\lambda^\ell\left[r_t+V(s_t)-\gamma V(s_{t+\ell}))\right]+V(s_t)\right\}$$

While this formulation appears complex, it comprises three key design elements:

- **Off-policy learning.** PPO optimizes the policy using data collected from an "old" policy $\pi_{\theta_{\text{old}}}$ (as seen in the expectation over $\pi_{\theta_{\text{old}}}$). However, the policy gradient theorem says that data should follow the distribution $\pi_\theta$ whenever we want to update $\pi_\theta$. Therefore, directly updating using data from $\pi_{\theta_{\text{old}}}$ would introduce bias. To eliminate this bias, PPO employs importance sampling (Kroese et al., 2013), represented by the ratio $\psi(s_t,a_t)$, which adjusts the data ratio to ensure it is unbiased in expectation.[1] This has a crucial practical implication: it enables *data reuse.* Even after updating policy to $\pi_\theta \neq \pi_{\theta_{\text{old}}}$, we can continue using data from $\pi_{\theta_{\text{old}}}$ rather than immediately collecting new data from $\pi_\theta$. This approach significantly reduces data collection requirements (Liang et al., 2025).

  In practice, implementing data reuse and importance sampling involves two key details: splitting the data into multiple mini-batches and training them over multiple epochs.[2] Traditional deep RL methods advocate multi-epoch training to maximize data efficiency (Schulman et al., 2017; Huang et al., 2022). However, in language model fine-tuning, single-epoch training is often preferred (Li et al., 2024c; Shao et al., 2024) due to the instability risks posed by repeated updates (Yao et al.,

---

[1]In the case of LLMs, there may be a mismatch between the training and inference policies, even when using the same parameters, due to differences in training precision and inference precision. This phenomenon, known as training-and-inference mismatch (Yao et al., 2025a; Liu et al., 2025a), introduces additional off-policy sources. To correct for this, importance sampling is required to ensure unbiased updates. We discuss this issue in greater detail in Section 4.

[2]For instance, the former corresponds to the `ppo_mini_batches` hyperparameter, while the latter aligns with `ppo_epochs` in the training framework `verl` (Sheng et al., 2024).

2023b). Even within single-epoch training, the utility of splitting data into mini-batches remains debated. An alternative approach is full-batch gradient descent for on-policy updates, eliminating importance sampling altogether. This was first demonstrated by (Li et al., 2024c) in the context of LLM post-training, with subsequent studies (He et al., 2025; Chen et al., 2025d) suggesting that on-policy updates enhance stability and mitigate entropy collapse.

We posit that these considerations fundamentally relate to the underlying data properties and offer theoretical insights from classical optimization learning theory (Bottou et al., 2018; Sun, 2019). Given a constant step size, splitting the data into mini-batches and performing Stochastic Gradient Descent (SGD) can converge faster than full-batch Gradient Descent (GD) when the data is *homogeneous*: in the extreme case where all samples are identical, SGD's gradient matches that of GD, but SGD processes $n$ times more samples than GD, where $n$ represents the total number of mini-batches. We conjecture that similar principles apply to RL optimization. If the data exhibits homogeneity, off-policy learning could prove more beneficial, provided identical learning rates are maintained. Conversely, with heterogeneous data, the advantages of off-policy learning may diminish substantially.

- **Clipping-based Regularization.** It is important to recognize that importance sampling, while effective, has inherent limitations: since samples from $\pi_{\theta_{\text{old}}}$ are finite and cannot encompass all possible scenarios, statistical estimation errors still exist even with importance sampling adjustments. To mitigate these errors, PPO implements probability ratios clipping as a mechanism to constrain excessive policy updates. Specifically, in the objective function above:
  - If $\widetilde{A}(s_t, a_t) > 0$ (positive advantage), the policy gradient naturally incentivizes increasing $\pi_\theta(a_t|s_t)$. However, when $\psi(s_t, a_t) > 1 + \delta$ (meaning $\pi_\theta(a_t|s_t) > (1+\delta)\pi_{\theta_{\text{old}}}(a_t|s_t)$), further updates yield zero gradient.
  - Similarly, if $\widetilde{A}(s_t, a_t) < 0$ (negative advantage), the policy gradient incentivizes decreasing $\pi_\theta(a_t|s_t)$, but the clipping mechanism prevents it from falling below $(1-\delta)\pi_{\theta_{\text{old}}}(a_t|s_t)$.

As explained in (Schulman et al., 2017), this functions as a single-sided (applying separately to either positive or negative advantage) and point-wise (affecting individual elements in the distribution) regularization mechanism. This approach contrasts with the holistic and distributional KL regularization employed in TRPO (Schulman et al., 2015). Furthermore, PPO's regularization is post-hoc (applied after the policy probability ratio exceeds the threshold), while TRPO's KL-regularization is preventative (ensuring the updated policy never exceeds the boundary).

Recent advances in the literature suggest that relative clipping may not be the optimal formulation. Two specific scenarios illustrate this limitation. First, in exploration scenarios involving potentially beneficial actions with initially low probabilities, it may be advantageous to substantially increase these probabilities to exploit discovered rewards. Addressing this, Yu et al. (2025b) introduced asymmetric regularization coefficients with a larger $\delta$ value for positive advantage cases, allowing for more aggressive probability increases when warranted. Second, excessively increasing the probability of already high-likelihood actions with positive advantages (i.e., too much exploitation) can undermine exploration of other potentially beneficial actions. To address this concern, Li et al. (2025g) developed ranking-based probability flow regularization, operating on the insight that maintaining relative ranking within the distribution, rather than driving probabilities to extreme values, is sufficient to preserve action diversity while still enabling effective learning.

- **Generalized Advantage Estimation.** PPO employs an advanced method, Generalized Advantage Estimation (GAE) (Schulman et al., 2016), for estimating advantage, which represents a "centralized" version of reward; see Equation (11). We first clarify that this centralization preserves the optimal action ordering, thus maintaining unbiasedness. Furthermore, while centralization alone does not alter variance, when combined with stochastic sampling, it effectively reduces variance, as analyzed in (Dayan, 1991; Li et al., 2024c).

The advantage calculation in PPO implements an Exponential Moving Average (EMA) of one-step advantage estimations. Specifically, the one-step advantage is computed as the difference between the immediate reward plus the discounted value at the subsequent state $V(s_{t+1})$ and the estimated value at the current state $V(s_t)$. The value function, defined as the expected long-term cumulative future

rewards, must be *separately trained* using Temporal Difference (TD) learning methods (Sutton, 1988). As Li et al. (2024c) point out, this separate training effectively doubles the required computational resources and significantly extends the overall training time. The intricacies of value network learning extend beyond the scope of this survey; interested readers are directed to (Sutton & Barto, 2018) for a comprehensive treatment of this subject. Finally, we note two critical hyperparameters govern the calculation of GAE: $\gamma \in [0, 1]$ controls the discount rate for future rewards, while $\lambda \in [0, 1]$ regulates the discount rate for future advantages. We will examine these hyperparameters later.

To summarize, PPO operates iteratively through the following process: 1) collecting trajectories using the current policy; 2) estimating advantages using the value function; 3) updating the policy using the clipped surrogate objective with SGD-style algorithms (usually Adam (Kingma & Ba, 2014)); 4) simultaneously training the value function to better estimate returns for the sampled trajectories. This process continues until convergence, demonstrating PPO's online nature.

We clarify that the algorithms introduced later mainly differ from PPO in their advantage estimation, while the first two ideas are directly applicable to these algorithms. We further expand on the third point below. We observe that the best practices for applying PPO to LLMs typically involve setting $\gamma, \lambda = 1$ (Yao et al., 2023b; Ahmadian et al., 2024). Under the further assumption of on-policy update, PPO effectively reduces to the REINFORCE estimator (Williams, 1992) with a baseline value (a point we will formally elaborate on later), where $V(s_t)$ is the baseline value (Li et al., 2024c). To put it in the context of LLM with outcome reward, PPO essentially reduces to

$$\mathcal{L}_{\mathrm{ppo}}(\theta) = \mathbb{E}_{x \sim \rho} \mathbb{E}_{y_{1:T} \sim \pi_\theta} \left[ \sum_{t=1}^{T} r(x, y_{1:T}) - V(x, y_{1:t}) \right].$$

This simplification reveals an important insight for LLM applications: the value network in PPO primarily provides a token-level baseline for gradient estimation. While this enables fine-grained credit assignment, it requires accurate value estimation, which remains challenging in practice. Notably, GAE and TD learning—central to traditional RL—play no role in this setting.

> **Takeaways: PPO reduces to REINFORCE with Token-wise Baseline**
>
> - PPO requires training two separate networks: the language model itself and a value network, which effectively doubles both the computational resources and training time.
> - When training LLMs with trajectory-level or outcome-based rewards, the value network primarily serves to provide a baseline for advantage estimation.

**From PPO to REINFORCE.** Recent research has focused on developing more efficient algorithms for LLM training. Li et al. (2024c) were the first to argue that PPO is computationally expensive for LLMs and developed methods that leverage the unique properties of language models. They highlighted that PPO was originally designed for general RL tasks where the value network serves a crucial purpose: in traditional RL environments, stochastic transitions create uncertainty beyond the agent's control, resulting in noisy return estimates. However, token generation in language models is deterministic once the prompt and model parameters are fixed (assuming greedy decoding). This fundamental difference implies that the "environment"—the LLM itself—does not introduce epistemic uncertainty. Consequently, the variance of direct policy gradients with Monte Carlo estimation can be significantly reduced, making REINFORCE-style methods viable.

Beyond this theoretical justification, eliminating the value model offers significant computational advantages. The GPU memory traditionally allocated to the value network can be reallocated to accelerate response sampling during exploration or to accommodate larger models and batches. Furthermore, the substantial training time and computational overhead associated with learning and updating the value network are completely eliminated. Specifically, ReMax reduces GPU memory consumption by approximately 46% and achieves roughly 1.6x faster training compared to PPO (Li et al., 2024c). Importantly, their research demonstrated that performance comparable to PPO could be achieved without a value network by building upon the simpler REINFORCE algorithm—a finding that would be highly improbable in classical deep

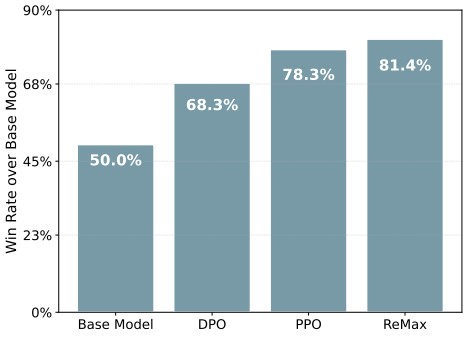
(a) Alignment task (Li et al., 2024c)

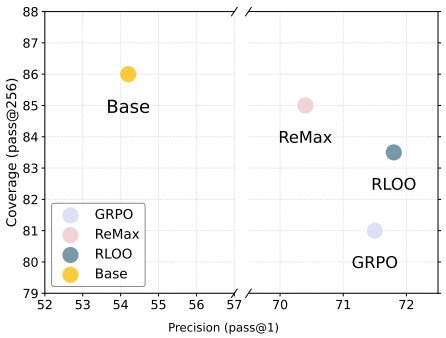
(b) Math reasoning task (Kruszewski et al., 2025)

Figure 7: Comparison of RL algorithms for LLM training. PPO, ReMax, GRPO, and RLOO achieve comparable performance on (a) alignment tasks and (b) math reasoning tasks, demonstrating that REINFORCE-based methods have strong performance, even matching PPO's effectiveness in the LLM setting.

RL applications due to REINFORCE's historically high variance. Recent studies have corroborated this observation (Kruszewski et al., 2025), with comparative results presented in Figure 7.

While this shift to simpler frameworks is promising, vanilla REINFORCE does not work directly (Li et al., 2024c). This is due to the reward heterogeneity issue arising from the multi-task nature of LLM training (Li et al., 2024c). Mitigating reward heterogeneity requires a prompt-dependent baseline value for normalization. This is precisely the REINFORCE with Baseline framework (Dayan, 1991; Sutton & Barto, 2018):

$$\widetilde{g}_{\text{REINFORCE-with-baseline}}(\theta) = \sum_{i=1}^{N} \left( r(x^i, y_{1:T}^i) - b^i \right) \cdot \nabla_\theta \log \pi_\theta(y_{1:T}^i | x^i) \quad \text{with } x^i \sim \rho, y_{1:T}^i \sim \pi_\theta(\cdot | x^i). \quad (12)$$

Here $b^i$ represents the baseline value to be designed. That is, we introduce a baseline value for reference and the term $r(x^i, y_{1:T}^i) - b^i$ is also often referred to as "advantage". Thus, a reference point is introduced. A key property is that this estimator remains unbiased, as demonstrated in (Williams, 1992). Next, we review recent advances in designing baseline values.

Most modern methods rely on direct Monte Carlo estimation, in contrast to the model-based estimation approach used in PPO. For clear comparison and understanding, we present the general form below:[3]

$$\widetilde{g}_{\text{REINFORCE-with-baseline}}(\theta) = \sum_{i=1}^{N} \sum_{j=1}^{K} \sum_{t=1}^{T} \nabla_\theta \log \pi_\theta(y_t^{[i,j]} | x^{[i]}, y_{1:t-1}^{[i,j]}) \cdot \texttt{advantage}(x^{[i]}, y_{1:t-1}^{[i,j]})$$

Here $K$ corresponds to the number of generated responses per prompt. For PPO, we have:

$$\texttt{advantage}_{\text{PPO}}(x^{[i]}, y_{1:t}^{[i,j]}) = r(x^{[i]}, y_{1:T}^{[i,j]}) - b_{\text{PPO}}(x^{[i]}, y_{1:t}^{[i,j]})$$

$$b_{\text{PPO}}(x^{[i]}, y_{1:t}^{[i,j]}) = V(x^{[i]}, y_{1:t}^{[i,j]})$$

Note that the baseline value is calculated at the token level. This approach can be beneficial when the value estimation is accurate (i.e., when the value network is well-trained (Yuan et al., 2025b)), but may be detrimental when the value network is imperfectly trained.

**Variants of REINFORCE.** Several key design approaches have emerged (ref to Table 3 for a summary):

- ReMax (Li et al., 2024c) selects the baseline value to the reward of greedy decoding response.

$$\texttt{advantage}_{\text{ReMax}}(x^{[i]}, y_{1:t}^{[i,j]}) = r(x^{[i]}, y_{1:T}^{[i,j]}) - b_{\text{ReMax}}(x^{[i]}) \quad (13)$$

$$b_{\text{ReMax}}(x^{[i]}) = r(x^{[i]}, \bar{y}_{1:T}), \text{ with } \bar{y}_t^{[i]} \in \operatorname{argmax} \pi_\theta(\cdot | x^{[i]}, \bar{y}_{1:t}^{[i]}) \quad (14)$$

---

[3]We omit techniques such as off-policy learning and clipping regularization for clarity of presentation, though these can be implemented in practice.

Table 3: Comparison of online RL algorithms for fine-tuning of LLMs. "Computational Overhead" refers to the extra cost beyond standard language model training, such as maintaining and training auxiliary models. *PPO's gradient is unbiased under on-policy updates but introduces bias with off-policy data reuse, which is a standard practice. †GRPO's bias stems from two sources: using the same samples to compute the mean baseline and to normalize by the standard deviation.

| Algorithm | Baseline Estimation | Computational Overhead | Gradient Bias | Gradient Variance |
|---|---|---|---|---|
| PPO | Value Model | High | Unbiased* | Low |
| REINFORCE | None | None | Unbiased | High |
| ReMax | Greedy Decoding | Low | Unbiased | Low |
| GRPO | Sample Mean | Low | Biased† | Low |
| RLOO | Leave-One-Out Mean | Low | Unbiased | Low |

The rationale is that greedy decoding represents the mode (i.e., the highest probability) of the policy distribution, which provides compact information and is computationally efficient.

- GRPO (Shao et al., 2024) replaces the value network estimation with the direct reward value.

$$\texttt{advantage}_{\text{GRPO}}(x^{[i]}, y^{[i,j]}_{1:t}) = \frac{r(x^{[i]}, y^{[i,j]}_{1:T}) - b_{\text{GRPO}}(x^{[i]})}{\text{std}\{r(x^{[i]}, y^{[i,1]}_{1:T}), \dots, r(x^{[i]}, y^{[i,K]}_{1:T})\} + \epsilon}$$

$$b_{\text{GRPO}}(x^{[i]}) = \text{mean}\{r(x^{[i,1]}, y^{[i,1]}_{1:T}), \dots, r(x^{[i,K]}, y^{[i,K]}_{1:T})\}$$

where $\epsilon$ is a small positive number (e.g., $10^{-6}$) to avoid numerical division error. We note that this baseline value is actually depeds on both $x^{[i]}$ and $\{r(x^{(i,1)}, y^{(i,1)}_{1:T}), \dots, r(x^{(i,K)}, y^{(i,K)}_{1:T})\}$, but we omit the later in the notation to emphasize that the baseline value is at the response-level and to avoid the notation cluster. This normalization is in part follows the normalization of advantage used in OpenAI's PPO implemention[4], with the exception that the normalization is done per prompt rather than across whole dataset.

- RLOO: Similar to GRPO, Ahmadian et al. (2024) leveraged the leave-one-out estimator in (Kool et al., 2019):

$$\texttt{advantage}_{\text{RLOO}}(x^{[i]}, y^{[i,j]}_{1:t}) = r(x^{[i]}, y^{[i,j]}_{1:T}) - b_{\text{RLOO}}(x^{[i]})$$

$$b_{\text{RLOO}}(x^{[i]}) = \frac{1}{K-1} \sum_{j=1:j\neq i}^{K} r(x^{[i]}, y^{[i,j]}_{1:T})$$

This estimator leverages the leave-one-out principle, which calculates the baseline for each response by averaging the rewards of all other responses generated for the same prompt.

One notable feature of the above baselines is that ReMax leverages only one response for baseline estimation, while others use multiple responses for baseline estimation. This is largely because when ReMax was proposed, the training infrastructure (DeepSpeed-Chat (Yao et al., 2023b)) did not support efficient sampling of multiple responses (actually supporting only 1 or 2 responses at that time), so using the greedy baseline value was the most efficient and effective approach in the context of limited sampling.

We note that the primary benefit of introducing a baseline lies in its variance reduction properties for stochastic gradients. Rigorous theoretical work by Dayan (1991) and Li et al. (2024c) demonstrates that appropriate baselines reduce gradient estimation variance. The underlying mechanism involves introducing a random variable $\nabla_\theta \log \pi_\theta(a_{1:T}|x) \cdot b(x)$ that positively correlates with $\nabla_\theta \log \pi_\theta(a_{1:T}|x) \cdot r(x, a_{1:T})$. This correlation reduces the second moment and consequently the variance—a technique known in statistical theory as a control variate (Kroese et al., 2013).

While alternative variance-reduction designs exist beyond REINFORCE with baseline, such as stochastic variance reduced gradient (Johnson & Zhang, 2013) approaches (demonstrated in (Papini et al., 2018) and (Xu et al., 2019)), these methods typically introduce computational overhead that exceeds the REINFORCE

---

[4]https://github.com/openai/baselines/blob/master/baselines/ppo2/model.py#L139

with baseline framework. Consequently, despite their theoretical merits, these alternative techniques have not seen widespread practical adoption.

We also note that unlike ReMax and RLOO that can be proved to be unbiased, GRPO's gradient estimator has inherent biases. There are two distinct error sources. First, the standard deviation calculation introduces statistical correlation because it uses the same random samples, resulting in bias. Recent work by Liu et al. (2025f) has proposed eliminating the standard deviation component altogether to address these potential bias issues in GRPO. Second, the mean estimation introduces statistical bias, because it depends on random samples $(x^{[i]}, y^{[i,j]})$. However, Kool et al. (2019) proved that this bias is largely eliminated (up to a scaling constant) when applying double summation over $j$. We first clarify that the bias does not mean that this algorithm cannot converge. From the perspective of optimization theory, as long as the angle between update direction and the gradient direction is smaller than 90 degrees, the algorithm can be expected to converge asymptotically. We also note that from the perspective of optimization algorithms, the standard deviation correction may be viewed as "adaptive learning rates" with the insight that a smaller learning rate should be employed for high-variance stochastic gradients (see, e.g., the formal theoretical arguments in (Bottou et al., 2018)). In the context of GRPO, this means that GRPO uses the reward variance (a simple statistic as a proxy for reward noise) and adaptively adjusts the learning rate for each prompt. Thus, the bias introduced here could alleviate the variance issue and in some cases could be beneficial (Liu et al., 2025g).

**The Optimal Baseline Value.** Since we aim to minimize the variance of stochastic gradients, a natural question arises: what is the best design for the baseline value in terms of variance reduction?

**Proposition 1** ((Greensmith et al., 2004; Li et al., 2024c))**.** *The minimal-variance baseline value for Equation* (12) *is*

$$b^\star(x) = \frac{\mathbb{E}_{y_{1:T} \sim \pi_\theta}[r(x, y_{1:T}) \cdot \|\nabla_\theta \log \pi_\theta(y_{1:T}|x)\|_2^2]}{\mathbb{E}_{a_{1:T} \sim \pi_\theta}[r(x, y_{1:T})]}.$$

This optimal baseline value incorporates the score function $\nabla_\theta \log \pi_\theta(y_{1:T}|x)$ into its calculation, as this component—in conjunction with the advantage function—directly influences gradient magnitude. Mechanistically, it uses the squared norm of the score function to re-weight reward values. However, this optimal baseline is computationally expensive to calculate, requiring additional forward and backward passes through the model. Consequently, there has been limited empirical investigation since it was first proposed. Recent work by Hao et al. (2025) has empirically studied this baseline value using various approximation techniques to make it more tractable in practice.

From a theoretically rigorous perspective, it is important to note that only the optimal baseline value achieves global variance reduction (remaining effective across any policy $\pi_\theta$), while baseline values employed in PPO, ReMax, GRPO, and RLOO achieve only local variance reduction. This fundamental distinction should be credited to the seminal work of Dayan (1991). For comprehensive theoretical developments on variance reduction techniques, readers are directed to both Dayan (1991) and the recent advances in Li et al. (2024c).[5]

Finally, we note that there are variants of policy optimization algorithms (Richemond et al., 2024; Gao et al., 2024; Kimi et al., 2025) that are derived not from the inner product objective of reward maximization as in Equation (4), but from other divergence functions. However, they share essentially the same insights as the above methods, so we do not provide detailed discussion here. Furthermore, there are emerging variants (see e.g., (Yu et al., 2025b; Yuan et al., 2025b; Chen et al., 2025b; Chu et al., 2025; Arnal et al., 2025; Zhang et al., 2025b)) of REINFORCE methods for LLM fine-tuning within specific contexts. To name a few, Hu et al. (2025) proposed global advantage normalization to improve training stability, while Zheng et al. (2025a) proposed group sequence policy optimization, which performs sequence-level importance-sampling-ratio correction to address training instability when training large-scale mixture-of-experts models. Additionally, GMPO (Zhao et al., 2025b) stabilizes training by optimizing the geometric mean of token rewards, and PKPO (Walder & Karkhanis, 2025) optimizes for pass@k performance to enhance sample diversity.

---

[5]To avoid potential misunderstandings, we emphasize that local variance reduction does not compromise algorithmic convergence. Rather, it indicates that under specific circumstances, the variance of REINFORCE-with-baseline might exceed that of naive REINFORCE. However, in such regimes the absolute variance is typically already sufficiently small, making relative comparisons between methods inconsequential for overall performance.

**The Dynamics of Policy Optimization.** While our previous discussion centered on the global design of policy gradient estimators—using techniques like baselines to reduce variance—we now shift our focus to the granular dynamics of the learning process. This section examines the functional role of individual tokens and their interaction with policy entropy. A growing body of research suggests that understanding these token-level dynamics is crucial for effective policy optimization (Li et al., 2025g; Wang et al., 2025c; Zhu et al., 2025b; Cui et al., 2025b; Yang et al., 2025d).

We first discuss the influence of high- and low-probability tokens. Recent studies highlight a fundamental tension in policy optimization. On one hand, maintaining high policy entropy is strongly correlated with better performance and exploration. For instance, Cui et al. (2025b) show that an LLM's exploratory capacity, quantifiable by its entropy, is a consistent indicator of success. Similarly, Wang et al. (2025c) reveal that tokens with high entropy are pivotal for guiding the model toward diverse reasoning paths. On the other hand, the optimization process can be disproportionately influenced by specific tokens. Yang et al. (2025d) find that low-probability tokens can dominate training due to their large gradient magnitudes. This creates a challenge: the very tokens that represent exploration can also destabilize learning. These findings collectively imply that training dynamics are critically sensitive to the gradients of certain tokens.

To better understand this phenomenon, we can visualize the interplay between a token's probability and the direction of its gradient update. As illustrated in Figure 8, this dynamic can be mapped onto a four-quadrant grid defined by two axes: *token probability* (low vs. high) and *gradient direction* (positive for reinforcement vs. negative for correction). Each quadrant represents a distinct learning scenario with a unique impact on policy entropy:

- Top-Left (Reinforcing a High-Probability Token): When a confident, correct prediction receives a positive gradient, the model's existing beliefs are reinforced. This sharpens the probability distribution and decreases entropy, potentially leading to overfitting—a scenario where regularization techniques like those in (Li et al., 2025g) become essential.

- Top-Right (Correcting a High-Probability Token): When a confident but incorrect prediction receives a negative gradient, the model is forced to correct a common mistake. This redistributes probability mass away from the incorrect token, naturally increasing entropy and encouraging exploration.

- Bottom-Left (Reinforcing a Low-Probability Token): In this key exploratory scenario, a novel action (a low-probability token) is rewarded with a positive gradient. This validates a new reasoning path and increases entropy by boosting a previously overlooked option. Successfully learning in this regime may require relaxing gradient update restrictions, as proposed by (Yu et al., 2025b).

- Bottom-Right (Correcting a Low-Probability Token): When an exploratory mistake receives a negative gradient, the model learns to prune an unproductive reasoning path. This removes a poor option from consideration and decreases entropy.

Crucially, in both scenarios involving low-probability tokens (the left two quadrants), the gradients have a large magnitude (Li, 2025). This makes their optimization a delicate balancing act, requiring careful management to harness the benefits of exploration without destabilizing the training process.

## 3.2 Offline Algorithms

This subsection reviews preference-based offline optimization methods for LLMs, represented by Direct Preference Optimization (DPO) (Rafailov et al., 2023), which updates the policy from a fixed preference dataset. This use of "offline" should be distinguished from classical offline RL, which typically learns from logged trajectories in generic decision-making problems and addresses distribution shift and over-estimation issues (Fujimoto et al., 2019; Kumar et al., 2020). Because classical offline RL has not yet been widely adopted for large-scale LLM post-training, we do not review it as a main algorithm family; readers may refer to (Levine et al., 2020; Prudencio et al., 2023).

**DPO as a representative offline preference-optimization method.** A representative work is Direct Preference Optimization (DPO) (Rafailov et al., 2023), which updates the LLM with a fixed preference dataset. Compared with the general RL framework, DPO solves a specific class of KL-regularized reward-maximization

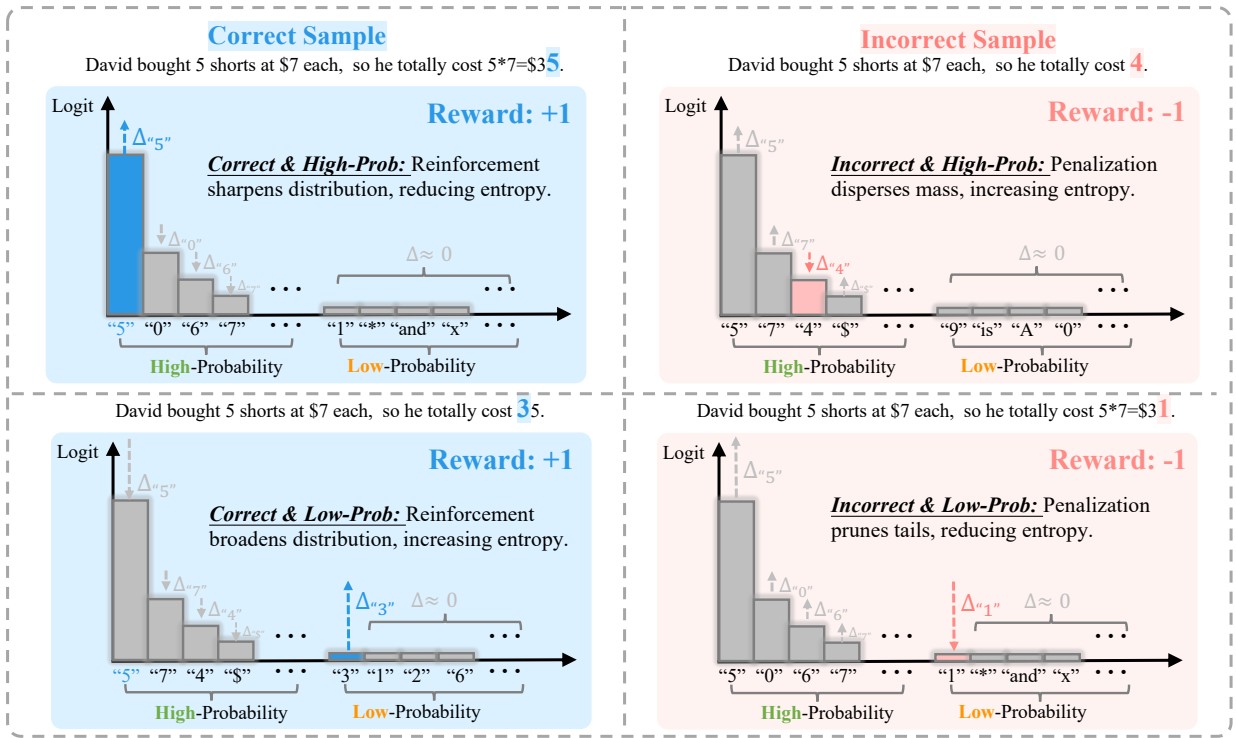

Figure 8: The four quadrants of gradient dynamics in policy optimization. This figure illustrates how token updates affect policy entropy based on the token's initial probability (x-axis) and the direction of the gradient (y-axis). Updates that increase entropy (fostering exploration) occur when correcting a high-probability token or reinforcing a low-probability one. Conversely, updates that decrease entropy (promoting exploitation) occur when reinforcing a high-probability token or correcting a low-probability one.

problems:

$$\pi^{\star} \in \underset{\pi}{\arg\max}\, \mathbb{E}_{y \sim \pi(\cdot|x)}[r(x,y)] - \beta D_{\mathrm{KL}}(\pi, \pi_{\mathrm{ref}}).$$

Here $\beta > 0$ is a hyperparameter controlling the strength of KL regularization, and $\pi_{\mathrm{ref}}$ is a reference model (usually the base model or SFT model). By introducing the KL regularization, there is a closed-form solution:

$$\pi^{\star}(y|x) = \frac{1}{Z(x)} \pi_{\mathrm{ref}}(y|x) \exp\left(\frac{r(x,y)}{\beta}\right).$$

where $Z(x) = \sum_y \pi_{\mathrm{ref}}(y|x) \exp(r(x,y)/\beta)$. This allows a one-to-one mapping between the reward function and the optimal policy $\pi^{\star}$. Hence, we can parameterize $\pi^{\star}$ by directly parameterizing $r$. This insight is leveraged in (Rafailov et al., 2023). They studied preference learning from the Bradley-Terry model loss, and the policy optimization objective can be derived from the closed-form solution:

$$\mathcal{L}_{\mathrm{DPO}}\left(\pi_\theta; \pi_{\mathrm{ref}}\right) = -\mathbb{E}_{(x,y_w,y_l)\sim\mathcal{D}}\left[\log\sigma\left(\beta\log\frac{\pi_\theta\left(y_w \mid x\right)}{\pi_{\mathrm{ref}}\left(y_w \mid x\right)} - \beta\log\frac{\pi_\theta\left(y_l \mid x\right)}{\pi_{\mathrm{ref}}\left(y_l \mid x\right)}\right)\right] \tag{15}$$

The key insight is that LLMs are viewed as implicit reward models. For preference data, the model's own log probabilities can be interpreted as indicators of how preferable one response is over another. During policy optimization, DPO directly adjusts the log-likelihoods to increase the probability of preferred responses relative to depreferred ones.

DPO simplifies the training pipeline by eliminating the need for an explicit reward model, making it appealing for practical applications. For example, LLaMA 3 (Grattafiori et al., 2024) replaces traditional RLHF with DPO optimization. In practice, various implementation tricks are often employed to improve stability and performance (Park et al., 2024; Pang et al., 2024; Ethayarajh et al., 2024; Wu et al., 2024a; Meng et al., 2024; Pal et al., 2024; Kim et al., 2025a; Lou et al., 2025). For instance, Park et al. (2024) studied the

length bias issue of DPO and proposed a length-normalized variant. Pang et al. (2024) observed that the likelihood of positive responses can decrease and proposed adding a next-token-prediction loss to mitigate this issue. SimPO (Meng et al., 2024) incorporates the average log probability, effectively aligning with model generation while removing the dependency on a reference model, thereby improving computational and memory efficiency. Xiao et al. (2024a) pointed out that RLHF algorithms, including DPO, cannot recover the true preference policy and proposed entropy regularization to address this limitation.

**Variants Beyond DPO.**  Several notable variants have been developed to address specific limitations of DPO. Identity Preference Optimization (IPO) (Azar et al., 2024) introduces a general theoretical framework ($\Psi$PO) that unifies existing RLHF and DPO methods. By setting $\Psi$ to the identity function, IPO directly optimizes preferences without relying on reward modeling or the Bradley-Terry assumption, thereby addressing the overfitting issues observed in DPO—particularly the problem of reward divergence when training for multiple epochs. Kahneman-Tversky Optimization (KTO) (Ethayarajh et al., 2024) takes a fundamentally different approach by leveraging insights from prospect theory. Unlike DPO and IPO which require paired preference data, KTO operates on unpaired binary feedback indicating whether an output is desirable or undesirable. This significantly reduces data collection costs since obtaining binary labels (thumbs-up/thumbs-down) is considerably easier than collecting pairwise comparisons. KTO has been shown to match or exceed DPO performance across model scales from 1B to 30B parameters.

Since the introduction of DPO, there has been increasing interest in extending it beyond the Bradley–Terry framework, as it cannot address the issue of conflicting preferences. Please refer to (Azar et al., 2024; Wu et al., 2024b; Swamy et al., 2024; Liu et al., 2025b) for recent advances and related work.

**Connection with Online RL Algorithms.**  Offline RL algorithms differ fundamentally from online RL algorithms (Li et al., 2024b; Nika et al., 2024). Xu et al. (2024) have presented comprehensive empirical results to support this claim. The core advantage of online RL algorithms lies in their "online optimization" characteristic, meaning the model can generate new responses and learn from this new data produced by the current policy. This contrasts with DPO's reliance on a human-crafted static dataset. The work by (Li et al., 2024b) emphasizes that the improvement potential of online RL algorithms is best realized when online updates are sufficient, including an adequate supply of model-generated prompts and responses. This implies that offline methods could be enhanced by introducing some form of "online" data collection and utilization mechanism, even if simulated or model-driven. For instance, iterative DPO (Xiong et al., 2023; Pang et al., 2024) generates new responses using the current model and then uses a learned reward model to label this new data for the next round of DPO training. This process introduces elements analogous to online learning, wherein the model learns from data generated by its own (or an updated) policy, thereby partially bridging purely offline learning with the dynamics of online learning and exhibiting some robustness to OOD issues.

**When to Prefer Offline vs. Online Methods.**  The choice between offline and online algorithms involves several practical considerations. A useful practical criterion is that preference-optimization or offline methods are often preferable when high-quality preference data or logged rollouts already cover the target distribution and infrastructure simplicity is important, whereas online RL becomes more attractive when rewards are cheaply verifiable and the policy must improve beyond a static dataset. Offline algorithms like DPO gained significant popularity during 2023–2024, partly due to their compatibility with existing pre-training and SFT infrastructure—they use standard gradient descent without requiring the complex multi-model setups (actor, critic, reward model) needed for PPO-based RLHF (Ouyang et al., 2022). This infrastructure compatibility substantially lowers the barrier to adoption. However, as RL infrastructure for LLMs has matured, online algorithms such as ReMax and GRPO have become increasingly popular, particularly for reasoning tasks where verifiable rewards are available. In the context of RLVR, online algorithms also offer computational advantages: for ReMax/GRPO with $m$ positive and $n$ negative samples, the gradient computation complexity is $O(m + n)$, whereas DPO-style algorithms require constructing all pairwise combinations, resulting in $O(m \times n)$ complexity. Nevertheless, offline algorithms retain significant advantages when the offline dataset has sufficient coverage and size to adequately represent the target distribution. Finally, we note that offline methods may have great potential when considering experience data, which we discuss further in Section 5.

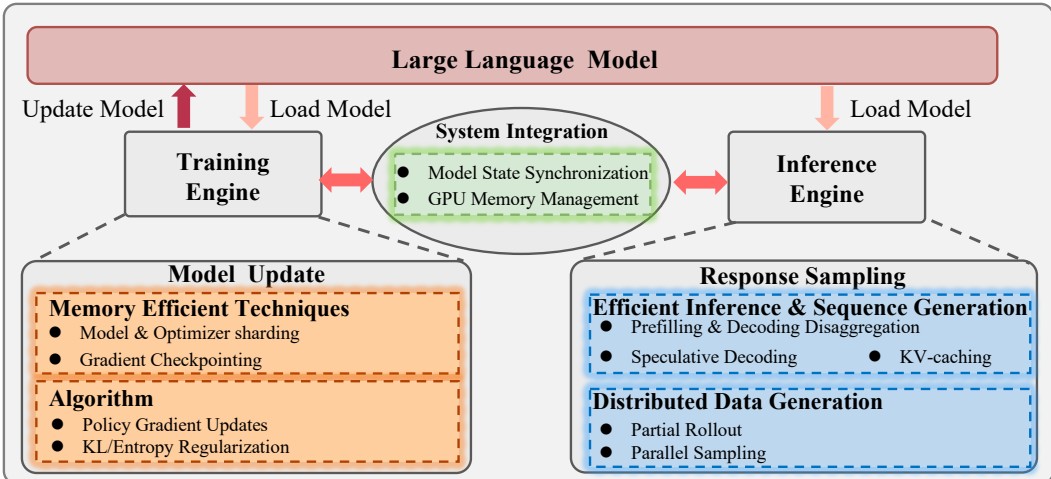

Figure 9: Architecture of a large-scale RL system for large language models. The framework separates the Model Update and Response Sampling pipelines, which are coordinated through a central System Integration module. The Model Update pipeline, executed by the Training Engine, refines model parameters via memory-efficient techniques (e.g., model and optimizer sharding, gradient checkpointing) and RL algorithms (e.g., policy-gradient updates, KL/entropy regularization). In parallel, the Response Sampling pipeline, executed by the Inference Engine, efficiently generates new data using methods such as prefilling and decoding disaggregation, speculative decoding, KV-caching, partial rollout, and parallel sampling. System Integration synchronizes model states and manages GPU memory across the training and inference loops, enabling continuous, low-latency data generation and model optimization.

## 4 Large-scale RL Training Framework

The preceding section focused on the algorithmic foundations of RL, particularly gradient estimation and optimization. These algorithmic advancements, when applied to LLMs, necessitate corresponding innovations in large-scale training infrastructure. We now transition from theory to practice, examining the significant systems engineering challenges required to support these algorithms. A primary challenge in implementing RL for LLMs stems from the need to concurrently support both model training and response sampling on GPUs. Specifically, while LLMs based on Transformers rely on KV-cache for accelerated inference (response sampling), this mechanism, designed for efficient generation, often introduces complexities when directly combined with the demands of model training on the same hardware. This inherent incompatibility in GPU resource management has recently spurred the development of new training infrastructure.

This section specifically focuses on practical system designs for the online RL framework presented in Algorithm 1, drawing from recent advancements in the field (Yao et al., 2023b; Xiao et al., 2023; Mei et al., 2024; Hu et al., 2024; Sheng et al., 2024; Shen et al., 2024; Fu et al., 2025; Wu et al., 2025; Wang et al., 2025d). Through an analysis of these techniques, we aim to provide practical guidance for implementing large-scale infrastructure and offer insights into improving the computational efficiency of RL systems, thereby enabling further scaling of RL applications for LLMs.

As mentioned before, RL training is an iterative process alternating between two key phases. The first is exploration, where the model generates diverse responses to find optimal solutions; from a systems perspective, this is the response sampling phase. The second is exploitation, where the model updates its parameters via gradient descent (i.e., backpropagation) to improve itself based on the collected data; this corresponds to the model updating phase. Accordingly, we will separately review response sampling and model updating before discussing their system integration.

## 4.1 Model Updating

In the algorithmic framework, the model update process is conceptually straightforward: model parameters are updated via gradient descent on a predefined loss function. In practice, however, implementation presents significant challenges. For instance, model sizes often exceed the memory capacity of a single GPU, while large datasets create computational bottlenecks that slow down training iterations.

Distributed training systems are the standard solution. Frameworks like FSDP (Zhao et al., 2023), DeepSpeed (Rajbhandari et al., 2020), and Megatron (Narayanan et al., 2021) overcome these hurdles using a combination of parallelism strategies. These strategies can be broadly categorized:

- Data Parallelism (Li et al., 2020) addresses large datasets by distributing data batches across multiple workers.

- Model Parallelism (e.g., tensor (Shoeybi et al., 2019) and pipeline (Narayanan et al., 2021) parallelism) addresses large models by partitioning the model itself—sharding components like MLPs and attention layers—across several GPUs.

We refer readers to existing surveys for a deeper technical dive (Verbraeken et al., 2020; Duan et al., 2024).

These strategies apply to pre-training, SFT and RL, since all involve gradient-based model updates. However, compared with SFT or pre-training on fixed datasets, RL training presents additional computational challenges. For instance, using mini-batches from a replay buffer creates an off-policy setting that requires careful hyperparameter design, particularly for learning rates (Hilton et al., 2022). Furthermore, properly setting the number of iterations or model updates for regularization schemes (e.g., KL-regularization) is challenging at scale (Gao et al., 2022). While pre-training addresses such issues through techniques like hyperparameter scaling laws (Bjorck et al., 2024; Li et al., 2025c; Kaplan et al., 2020) and $\mu$-Parameter transfer (Yang et al., 2022), efficient parameter setting in large-scale RL remains largely unexplored territory. We believe these research questions are practically important as we scale model and compute during RL training and offer substantial research opportunities.

## 4.2 Response Sampling

Beyond model updates, RL training requires continuous response sampling to explore and collect high-quality data. While conceptually straightforward—involving standard auto-regressive sampling—the practical implementation presents significant challenges. The interactive nature of RL demands low-latency response generation throughout training, which consumes substantial GPU memory and necessitates specialized acceleration techniques.

The computational complexity differences between sampling and training are particularly striking. During sampling, the auto-regressive nature of Transformers is inherently sequential, resulting in $\Theta(T)$ time complexity to generate $T$ tokens, as it requires $T$ separate forward passes. This contrasts sharply with model updates, where the gradients for all $T$ tokens can be computed in parallel with a single forward pass, achieving $\Theta(1)$ complexity in terms of passes. We review several key optimization directions below.

- **Caching and Memory Management.** KV caching is indispensable for accelerating generation, as it stores previously computed key–value pairs to avoid redundant computation (Pope et al., 2023). However, as memory demands scale, efficient cache management becomes increasingly critical. Modern inference systems such as vLLM (Kwon et al., 2023) and SGLang (Zheng et al., 2024b) employ techniques like paged attention (Kwon et al., 2023) to improve cache scheduling and utilization. Recent work further explores prefilling/decoding disaggregation (Qin et al., 2024; Chen et al., 2025c), which distributes prefilling and decoding across separate instances to enhance overall throughput.

- **Algorithmic Acceleration.** As context lengths grow, algorithmic improvements for inference become increasingly vital. Notable advances include speculative decoding (Chen et al., 2023; Xia et al., 2025), which leverages smaller models (or auxiliary prediction heads) to propose candidate tokens verified by larger models, and lookahead decoding (Fu et al., 2024), which reformulates

generation as a nonlinear system solved with Jacobian iteration, enabling simultaneous decoding of multiple tokens.

- **Improved Architectures.** To mitigate the sequential bottleneck of auto-regressive generation, researchers are exploring parallel sampling approaches inspired by diffusion models (Lou et al., 2023; Song et al., 2025; Deepmind, 2025). Within Transformer components, attention remains the primary computational bottleneck for long contexts, motivating more efficient designs. Examples include parameter-sharing methods like grouped query attention (Ainslie et al., 2023), dimensionality-reduction techniques such as multi-head latent attention (Liu et al., 2024), linear attention (Katharopoulos et al., 2020; Li et al., 2025a), looped Transformers (Giannou et al., 2023; Zhu et al., 2025a), and sparsity-aware approaches that exploit the inherent structure of attention weights (Yuan et al., 2025a; Lu et al., 2025).

Beyond module-specific optimizations, algorithm-hardware co-design approaches (Zhao et al., 2025a) exemplified by FlashAttention (Dao et al., 2022) leverage hardware properties such as memory hierarchy to achieve substantial speedups. We acknowledge that the above review covers only a subset of active research directions in inference acceleration. For comprehensive coverage of this rapidly evolving field, we refer readers to recent surveys (Zhou et al., 2024; Li et al., 2024a; Park et al., 2025).

## 4.3 System Integration

The previous sections examined techniques for model updating and response sampling in isolation, each employing distinct systems and optimization strategies. However, integrating these components into a unified large-scale system presents substantial coordination challenges.

**Centralized Orchestration.** Effective integration demands a centralized orchestration framework capable of scheduling model updates and response sampling while dynamically managing GPU memory allocation across the distributed system. Furthermore, it also manages the model parameter synchronization between training engine and inference engine. Modern implementations such as OpenRLHF (Hu et al., 2024), Verl (Sheng et al., 2024), and ReaL (Mei et al., 2024) demonstrate this architectural approach, providing coordinated resource management and workload distribution.

**Limitations of Synchronous Training.** The systems described above primarily employ synchronous training paradigms, where model updating and response sampling proceed in strict sequential order. This approach is sub-optimal in practice due to inherent variability in sampling times across different prompts, leading to resource underutilization and synchronization bottlenecks. The problem becomes particularly acute for complex tasks such as code generation, where reward computation may involve executing multiple unit tests, significantly extending the feedback loop and exacerbating idle GPU time (Luo et al., 2025).

**Asynchronous Training Paradigms.** To address these inefficiencies, asynchronous training architectures have emerged as a promising alternative, enabling partial rollouts, concurrent execution of response sampling and reward calculation (Fu et al., 2025; Kimi et al., 2025; Xia et al., 2025). By decoupling these processes, asynchronous systems can eliminate synchronization bubbles and achieve higher resource utilization.

**Reward Model MLOps and Reward Serving.** In integrated RLHF stacks, the reward model (or judge) becomes a continuously deployed, latency-sensitive service rather than a static component. In practice, the reward pipeline must handle: (i) iterative data collection and curation (preference logging, deduplication, adversarial filtering), (ii) versioned reward model training and evaluation, (iii) deployment/rollback of reward checkpoints, and (iv) monitoring for distribution shift and reward hacking as the policy changes over iterations (Dong et al., 2024; Wolf et al., 2025). Operationally, these constraints often push systems to decouple reward computation (e.g., remote RM services, caching, or staged filtering) from actor rollouts to preserve end-to-end throughput while maintaining guardrails against reward model drift (Hu et al., 2024; Dong et al., 2024).

**Training-Inference Mismatch.** As mentioned earlier, modern RL frameworks typically employ distinct computational engines for response sampling and model training—such as vLLM for efficient inference and Megatron or FSDP for distributed training—each optimized for their respective workloads. This architectural separation can lead to numerical discrepancies in token probability computations between the two engines, arising from differences in floating-point precision handling, kernel implementations, operator fusion strategies,

and computational graph optimizations (He & Lab, 2025; Yao et al., 2025b). Consequently, even models with identical parameters can produce different probability distributions, effectively transforming nominally on-policy algorithms into off-policy ones. These seemingly minor numerical differences can accumulate during the iterative RL process, causing divergence between the policy used to generate training data and the policy being evaluated during training. This divergence exacerbates critical issues such as catastrophic training collapse, particularly when low-probability tokens drive unstable gradient updates (Liu et al., 2025a). With truncated importance sampling, the downstream performance can be improved by about 5 points on 32B models (Yao et al., 2025b). Recent work further suggests that mitigating the mismatch can impose non-trivial systems overhead: e.g., truncated importance sampling requires an additional forward pass to obtain training-side log probabilities, increasing training compute by about ∼25% in a representative implementation (Qi et al., 2025). Complementary systems approaches aim to eliminate mismatch at the source by enforcing determinism across inference configurations; for instance, TP-invariant kernels can achieve bit-wise reproducibility and report zero probability divergence across tensor-parallel settings in controlled experiments (Zhang et al., 2025e).

**The Off-Policy Challenge.** Overall, both asynchronous training paradigms and training-inference mismatch create off-policy learning scenarios where the policy generating training data differs from the policy being updated, potentially leading to training instability and convergence issues. Addressing these challenges could draw from established techniques in classical RL literature, including methods for safe policy updates that bound deviation from reference policies (Munos et al., 2016), and divergence control mechanisms that prevent catastrophic policy degradation (Wang et al., 2019; Liang et al., 2025). The tension between computational efficiency and algorithmic stability in asynchronous RL systems represents a rapidly evolving research frontier, with significant opportunities for innovation spanning both system architecture and algorithmic design.

## 5 Discussion and Future Directions

Throughout this survey, we primarily examine the algorithmic and computational aspects of RL for LLMs. We delve into several important discussions not extensively covered in previous sections. The first topic is the foundational capabilities of LLMs trained by RL and their scaling laws. The second topic addresses the high-level features and objectives we aim to achieve through RL training. The final topic explores advanced capabilities that we seek to incorporate for real-world applications.

**What can RL actually teach a pretrained language model?** For well-pretrained LLMs, we know that RL can fine-tune models to enhance cognitive abilities such as chain-of-thought reasoning (Wei et al., 2022b), tool usage (Li et al., 2025e), and self-reflection in test-time scaling (OpenAI, 2024; Guo et al., 2025). Crucially, fine-tuning here refers to using a relatively small compute budget (e.g., less than 1%) compared with pre-training. Two fundamental questions emerge: Can we further scale this post-training compute, and what new behaviors can RL enable beyond pre-training alone?

Regarding this fundamental question, Zhou et al. (2023a) were among the first to investigate this issue and proposed the well-known "superficial alignment hypothesis." They argued that "a model's knowledge and capabilities are learned almost entirely during pre-training, while alignment teaches it which subdistribution of formats to use when interacting with users." Their compelling evidence came from demonstrating that a model trained via SFT with approximately 1,000 carefully structured samples could generate human-like responses—suggesting that models quickly learn stylistic conventions while drawing their substantive knowledge from pretraining (Lin et al., 2023a).

This hypothesis found further empirical support in mathematical reasoning domains. Studies such as DeepSeekMath (Shao et al., 2024) revealed an interesting pattern: while RL training did not increase the model's "ceiling performance" (defined by pass@K), meaning it could not solve fundamentally harder problems than before, it dramatically improved pass@1 rates, indicating that models became significantly better at selecting their optimal response on the first attempt. This suggested that rather than expanding raw problem-solving capabilities, RL functioned as a sophisticated confidence calibrator, teaching models to distill their best reasoning into their default response and making greedy generation more reliable. This phenomenon has been systematically investigated by Yue et al. (2025).

However, we note that these "negative results" have limitations, as indicated by emerging studies. First, treating pass@K based on final answer correctness as the sole metric of ceiling performance has inherent flaws. This is because a response may involve process errors (Zheng et al., 2024a) or yield the correct outcome merely by chance. Thus, pass@K could be overestimated when used to measure "ceiling performance." An interesting study by Wen et al. (2025) showed that RL can, in fact, improve fundamental capabilities. For instance, they demonstrated that RL not only improves outcome correctness but also reduces process errors, despite receiving only outcome-based rewards. Second, existing "negative results" often operate within a small compute regime. This "small" regime encompasses several factors: limited data size (usually 10K-100K samples), restricted exploration (sampling 8, 16, or 32 responses per prompt), and few training iterations (usually several epochs). Collectively, these factors suggest that current RL training paradigms are often insufficient. Several recent works (Liu et al., 2025d) have extensively expanded computational resources for RL and observed promising results. Despite this, we would like to remark that existing RL algorithms also have limitations in exploration, preventing the model's full capabilities from being unlocked. To illustrate, hard tasks (e.g., winning in the game of Go, writing a competition-level math solution, or solving an open question in a scientific domain) may require hundreds or thousands of trials and errors to find a correct solution. If limited sampling is used, the received rewards are often all negative, leading to zero training gradients. Furthermore, current infrastructure often does not adequately support such extensive exploration, thereby limiting the capabilities that RL can unlock. In summary, we expect enhanced RL to unlock new capabilities in the post-training process by interacting with environments and receiving feedback not covered by pre-training. This calls for algorithmic, training infrastructure, and evaluation advances in this direction.

**Algorithmic Features for Future Development.** Following the previous discussion, as RL algorithms become more stable and training infrastructure improves, several key directions emerge for algorithmic advancement. We highlight two critical areas:

- **Efficient and Effective Exploration.** Current RL methods for LLMs often suffer from inefficient exploration of the vast action space defined by token sequences. Consequently, the trials attempted may receive entirely negative outcome feedback, preventing the model from learning anything. To address this, extensive exploration strategies are crucial. Future algorithms must develop more advanced exploration techniques that can navigate the high-dimensional discrete space of language generation while maintaining training stability. This could be achieved through compact state/action space representations (Barrault et al., 2024; Jia et al., 2025), optimizing the allocation of the exploration budget (Li et al., 2025h), leveraging more structured exploration spaces (e.g., tree-based approaches) (Zheng et al., 2025c; Li et al., 2025f), or exploiting the unique properties of LLMs with hint-guided exploration techniques (Huang et al., 2025b; Zhang et al., 2025c).

- **Learning from Offline Data.** While online RL shows promise, the expensive cost of online exploration (both computational and monetary, due to large model sizes and slow simulation) makes it prohibitively expensive in practice. Furthermore, online RL algorithms often do not fully exploit the data they collect; for instance, algorithms like PPO and GRPO typically train on samples once and then discard them. Given that the online RL process can be viewed as a data collection process where mixed-quality data is accumulated, it becomes valuable to learn from this kind of experience data (Levine et al., 2020; Sinha et al., 2022; Li et al., 2023). There has been no extensive investigation of this approach in LLMs yet, with SFT (which typically involves imitation from high-quality, curated data) remaining dominant. However, leveraging such data will be crucial for future developments in the field (Silver & Sutton, 2025). Emergent work includes Zhang et al. (2025a), who proposed utilizing self-generated rollout trajectories in environments without explicit reward functions, training agents by predicting environmental transitions and analyzing their own decision-making errors.

- **Off-policy Learning.** Existing RL approaches for LLMs primarily rely on on-policy learning, where data is collected directly from the current policy. Breaking this limitation offers several significant advantages. First, as discussed, effectively learning from offline data inherently requires off-policy capabilities to maximize the utility of past experiences. Second, from a training system perspective, asynchronous training—which we previously noted can improve GPU utilization efficiency—naturally leads to off-policy data collection. While off-policy learning has historically introduced instability issues (Kumar et al., 2019; Hilton et al., 2022), these challenges could potentially be mitigated by

leveraging the unique characteristics of LLMs (Fu et al., 2025). For instance, Zheng et al. (2025b) introduced M2PO, which applies variance-based regularization on probability ratios to selectively filter training instabilities while retaining useful gradients, demonstrating that appropriately managed stale rollouts can match fresh data performance.

**Emerging Capabilities for Real-World Applications**  Pre-trained LLMs are primarily based on web text corpora and, as a result, cannot possibly encompass all scenarios encountered in physical and interactive environments, for which pre-training provides limited coverage. This scenario requires LLMs to leverage external tools and receive more external feedback, rendering RL increasingly necessary. We highlight several topics below.

- **Tool Usage and External Integration.** For complex tasks (such as fixing the issue in an Github repo (Jimenez et al., 2023)), solving problems solely within the token space can be both challenging and computationally prohibitive for LLMs. A promising strategy involves offloading specific subproblems to external tools, thereby abstracting low-level reasoning steps into tool calls and reducing the model's reasoning complexity (Shen, 2024; Jiang et al., 2024). This approach not only simplifies the training process but also enhances performance by mitigating hallucinations and producing more reliable outputs (Béchard & Ayala, 2024; Gao et al., 2023; Huynh & Lin, 2025). However, LLMs do not inherently learn effective tool invocation strategies through pretraining alone, especially when tool choice requires *API discovery/retrieval*, *schema-grounded argument generation*, and *post-call verification* under strict latency/cost budgets (Schick et al., 2023; Qin et al., 2023). Post-training techniques, particularly RL, offer a pathway to teach models efficient tool usage, enabling them to learn when and how to delegate subtasks appropriately. This requires LLMs not only to recognize their own limitations but also to strategically delegate subtasks to appropriate external tools or resources, often via modular agent architectures (e.g., planner–router/API retriever–executor with constrained decoding and error recovery) (Qin et al., 2023). Recent efforts have focused on training LLMs to interact with tools through long-chain-of-thought frameworks using RL, which help mitigate spurious or inefficient calls during complex problem-solving (Jin et al., 2025; Chen et al., 2025a; Li et al., 2025b;e; Zheng et al., 2025d; Xu & Peng, 2025). *Concrete open challenges* include: (i) long-horizon credit assignment across multi-tool trajectories (where intermediate tool outputs are non-differentiable and rewards are sparse); (ii) robust tool-call calibration (precision/recall of calls, retries, and fallback policies) under tool failures and non-stationary APIs; and (iii) standardized evaluation beyond single-tool settings, leveraging benchmarks that test realistic multi-step interactions and end-state correctness (e.g., ToolBench/ToolLLM, AgentBench, WebArena, and $\tau$-bench) (Qin et al., 2023; Liu et al., 2023b; Zhou et al., 2023b; Yao et al., 2024). How to leverage diverse tool ecosystems to solve more general and complex problems remains an open and important research question.

- **Multi-Agent Coordination and Collaboration.** The future of RL-enhanced LLMs extends beyond individual agent capabilities to multi-agent coordination. As multiple AI agents begin collaborating on complex tasks, RL becomes essential for learning effective communication protocols, task decomposition strategies, and coordination mechanisms. This includes learning how to negotiate, delegate responsibilities, resolve conflicts, and maintain coherent joint objectives across multiple interacting agents. Multi-agent RL for LLMs presents unique challenges, including the need for agents to develop shared communication languages, understand others' capabilities and limitations, and adapt their behavior based on the evolving dynamics of the agent collective (Cemri et al., 2025; Yang et al., 2025c; Tan et al., 2025; Tran et al., 2025). These capabilities are crucial for deploying LLM-based systems in real-world scenarios where multiple AI agents must work together or interact with human users in collaborative settings.

- **Multi-Modal RL and Physical World Interaction.** The integration of multi-modal capabilities represents a critical frontier for RL-enhanced LLMs (Huang et al., 2025a). Image and video understanding enable models to perceive and interact with the physical world in ways that pure text-based systems cannot achieve (Liu et al., 2023a; Bai et al., 2023). Audio and speech processing further enhance LLMs' ability to engage seamlessly with humans, fostering more natural and responsive interfaces (Zeng et al., 2024; Zhang et al., 2025d). Through RL training on multi-modal tasks,

LLMs can learn to ground their language understanding in visual and spatial contexts, developing more robust real-world reasoning capabilities (Yang et al., 2025b). *A central technical bottleneck is cross-modal reward alignment:* reward models must score helpfulness while explicitly penalizing ungrounded or hallucinated claims with respect to images/videos, and remain robust to reward hacking (Sun et al., 2024; Yu et al., 2024; Ahn et al., 2024). Multi-modal RL also raises *action space design* questions: actions may be discrete (language/tool actions), continuous (motor controls), or hierarchical combinations where an LLM proposes skill-level plans that low-level controllers execute (Ahn et al., 2022; Driess et al., 2023). Recent vision-language-action formulations further unify perception and control by tokenizing actions or generating action sequences autoregressively (Zitkovich et al., 2023; Jiang et al., 2022). Finally, evaluation should measure not only preference scores but also task success and generalization in embodied or interactive benchmarks with controlled splits (e.g., ALFWorld, VIMA) (Shridhar et al., 2020; Jiang et al., 2022). As this field rapidly evolves, advances in multi-modal RL continue to push the boundaries of embodied AI.

# 6 Conclusion

In this survey, we have traced the evolution of reinforcement learning for large language models across three key dimensions. We examined the unique algorithmic challenges posed by LLMs' discrete, high-dimensional nature and the specialized methods developed to address them. We explored the computational frameworks that have made large-scale RL training feasible, enabling the transition from proof-of-concept to production systems. Finally, we engaged with the fundamental debate about post-training's true capabilities: whether RL teaches new knowledge or refines the expression of existing knowledge, and how this understanding shapes our approach to capability development.

As we stand at this inflection point, the field faces exciting challenges around scaling laws, sample efficiency, and evaluation frameworks that capture RL's full impact. We hope this survey serves as both a reference for newcomers and a catalyst for existing researchers to explore the many open questions that remain. The story of RL for LLMs is still being written, and its most transformative chapters may well lie ahead.

# 7 Broader Impact Statement

Reinforcement learning has become a central tool for shaping LLM behavior, unlocking powerful capabilities while introducing meaningful risks that require careful consideration. We highlight three key concerns and constructive pathways forward.

**Reward Hacking and Safety.** Optimizing learned or engineered rewards can lead policies to exploit evaluator blind spots, achieving high scores without genuine improvements. In safety-critical settings, this may manifest as manipulative or deceptively persuasive behaviors (Gao et al., 2022; Mac Kim et al., 2025). *Mitigation strategies* include conservative optimization methods (e.g., KL regularization, reward ensembles), systematic red-teaming to identify failure modes, and complementary evaluations beyond training-time metrics to ensure robust generalization (Coste et al., 2023; Ganguli et al., 2022; Zou et al., 2023).

**Value Pluralism and Fairness.** RLHF optimizes for preferences drawn from specific annotator pools and institutional protocols, raising the "whose values?" question. This population-dependence means alignment reflects particular operationalizations rather than universal human values (Barnhart et al., 2025; Dahlgren Lindström et al., 2025). Furthermore, reward optimization can amplify systematic biases present in reward models or benchmarks (Lambert et al., 2024). *Best practices* include transparently reporting preference-collection details, stress-testing reward models under distribution shift, incorporating diverse stakeholder perspectives, and evaluating disparate impacts across demographic groups and topics.

**Capability Trade-offs and Resource Costs.** Post-training may reduce certain desirable abilities (creativity, long-tail coverage) while enhancing targeted objectives—an "alignment tax" (Ouyang et al., 2022; Lin et al., 2023b). Additionally, large-scale online RL substantially increases computational demands through repeated sampling and auxiliary model maintenance, contributing to energy consumption and environmental impact

(Patterson et al., 2021). *Responsible development* should track not only benchmark gains but also robustness metrics, distributional fairness, computational efficiency, and preserved capabilities across diverse contexts.

Addressing these challenges requires interdisciplinary collaboration among ML researchers, ethicists, policy-makers, and affected communities. Progress toward beneficial AI systems depends on technical innovation coupled with thoughtful consideration of societal implications and proactive risk mitigation.

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
