# OpenReview forum: "Review of Reinforcement Learning for Large Language Models: Formulations, Algorithms, and Opportunities"
_TMLR — Decision pending for TMLR_

### Review · Reviewer_Gd7K · 2025-10-27

**Summary Of Contributions:**

The paper provides a survey of RL methods for large language models, with particular focus on the finetuning step. It adresses the actual problem formulation, algorithms, and considerations that arise from the large scale of the underlying models. Furthermore, it discussed future directions like using verifyable rewards, multi-agent, and multi-modal approaches. Additionally, the work defines the 'four quadrants' of gradients dynamics in policy optimization, which from my perception is a small but interesting novel concept.

From my perception, the main contributions are as follows:
- Provide a concrete MDP formulation for training LLMs with sparse outcome-based rewards
- Compared advantages and drawbacks of PPO-based approaches compared to more simple REINFORCE setups (with baseline)
- Discusses offline vs. online learning
- Reviews practical requirements regarding infrastructure for training the resulting large-scale models

Key strengths:
- Good overview of recent progress in RL4LLM (quite complete from my perception)
- Clear discussion of why REINFORCE might be sufficient and no value functions need to be estimated
- Extensive coverage of methods for speeding up training and inference of large-scale RL

Key weaknesses (details below):
- Some of the claims might be formulated a bit too decisive
- Limited quantitative comparisons

**Additional Comments:**

N/A

**Audience:**

Yes

**Audience Explanation:**

I think this survey is a pretty good fit for TMLR, as it also uncovers new connections, is certainly a 'trendy' topic, and also discusses and rises new problems. It should be usefull and interesting for researchers that are working both in RL and LLMs.

**Broader Impact Concerns:**

I think the paper should at least put some emphasys on safety. Concepts like reward hacking are briefly mentioned, but given the importance of the topic also some methods to address these issues should be covered (our-of-distribution checks, adversarial evaluation, ...).

**Claims And Evidence:**

Yes

**Claims Explanation:**

I think the paper provides a quite comprehensive overview of relevant literature on the topic. The made claims (e.g. REINFORCE vs. PPO, offline vs. online RL) are substantiated in detail by fitting references.
Also the discussion regarding efficiency (FSDP, pagen attention, ...) seem to reflect up-to-date work.

**Requested Changes:**

From my perception there are no major caveats of the current work. However, there are a few points that would improve the quality of the work:
- A claim that should be discussed with a bit more care is the one on determinism and "no epistemic uncertainty" when discussing PPO vs. REINFORCE. The procedure itself still should have quite some uncertainty fro noise models, sampling strategies, off-policy data, etc.. It should be discussed how these affect the role of the baseline and value estimation.
- When discussing variants of REINFORCE in 3.1 I think the readability of the work would benefit from a small tabular overview of the different approaches (covering bias-variance properties, overhead, responses per prompt, ...)
- Concerning 3.2 "Connection with Online RL Algorithms": Here it would be interesting to have a short discussion hot to in principle re-use online trajectories (experience replay, constraint policy improvement with logged data) tailored for LLMs
- The work would benefit from a kind of 'notation table' at the beginning, summarizing the important notations ('s', 'x', 'y', 'a', etc.)

A few typos I stumbled upon:
- Bottom of page 3: 'minmize'
- Mid page 14: 'exaime'
- Figure 7: 'High-Proabability'

---

> ### Comment · Action_Editor_9dDd · 2026-03-17
> **To reviewer**
>
> Hi reviewer Gd7K,
>
> Would it be possible for you to answer the points made by the authors?
>
> Thanks,
>
> AC

---

> > ### Comment · Reviewer_Gd7K · 2026-03-17
> > **Updated Review**
> >
> > Hi Editor, Authors,
> >
> > Sorry for the delay, I must have missed the notification.
> >
> > I checked the new manuscript and my original review, almost all of the comments have been fully adressed.
> > Only two minor comments from my side:
> >
> > - Claim on 'no epistemic uncertainty': While this is sufficiently addressed / rephrased in the main parts of the paper, this statement still appears in the caption of Figure 6.
> > - There is still the 'proabability' typo, now in Figure 8.

---

> > > ### Author Response · Authors · 2026-03-23
> > >
> > > We sincerely thank you for carefully checking our paper! We apologize for overlooking these issues. They have now been corrected in the revised version. Specifically, we have clarified the "no epistemic uncertainty" phrasing in Figure 6 and fixed the typo in Figure 8.

---

> > > > ### Comment · Reviewer_Gd7K · 2026-03-23
> > > >
> > > > With that all my comments have been addressed, thanks a lot for the carefull consideration and re-working.

---

### Review · Reviewer_WcTz · 2025-11-22

**Summary Of Contributions:**

Survey examines the integration of RL techniques with LLMs, addressing a critical gap in AI alignment and post-training optimization.

The paper's core contributions include:
1. Theoretical Foundation: Establishes rigorous mapping between LLM generation processes and MDPs, providing formal frameworks for RLHF and RLVR paradigms. Unique insights include deterministic state transitions in LLMs and their algorithmic implications.
2. Algorithmic Analysis: Presents thesis that deterministic LLM transitions eliminate one key motivation for actor-critic methods (environmental stochasticity), enabling simpler REINFORCE-based algorithms to achieve competitive performance at lower computational cost.
3. Systems Engineering: Addresses large-scale implementation challenges including distributed training frameworks, memory optimization, and the "training-inference mismatch" problem that transforms nominally on-policy algorithms into off-policy ones.
4. Future Directions: Identifies key research opportunities in exploration efficiency, off-policy learning, tool integration, and multi-modal RL, grounded in discussion of the "Superficial Alignment Hypothesis" debate.

Strengths:
- Good timeliness, Mathematical rigor and comprehensive coverage
- Scholarly discussion of foundational debates (Superficial Alignment Hypothesis)
- Comprehensive treatment of challenges (model scale Figure 4, reward heterogeneity Figure 5)

Weaknesses(Addressed):

~~- Algorithmic performance claims (e.g., "comparable performance") need stronger empirical validation~~

~~- Limited process rewards coverage~~

~~- Missing quantitative analysis of computational trade-offs (wall-clock time, memory usage)~~
- Algorithmic performance claims now supported by Figure 7 empirical comparison
- Process rewards coverage expanded with dedicated subsection
- Quantitative analysis of computational trade-offs now included

**Audience:**

Yes

**Audience Explanation:**

This survey addresses one of the most commercially significant and academically active areas in machine learning. The interest is justified by:
1. Consolidation Value
2. Practical Relevance
3. Research Guidance
4. Bridge Between Communities
5. Timeliness

This survey addresses critical interests of multiple TMLR constituencies including LLM Post-Training Researchers, Agent Systems Developers, Systems Researchers, and Safety and Alignment Researchers.

Target audience segments:
- ML researchers entering LLM alignment
- Practitioners implementing policy optimization & RLHF/RLVR at scale
- Systems researchers optimizing training infrastructure

Anyone doing post-training optimization will find this useful. The practical guidance on when to prefer offline vs. online methods, and the MLOps considerations for reward model deployment, are the kind of details researchers need.

**Broader Impact Concerns:**

The paper addresses safety concerns throughout (reward hacking, adversarial attacks, alignment tax, red-teaming in Section 2.3). The treatment of reward hacking as a safety risk (not just a performance issue) and the acknowledgment that RLHF optimizes for specific populations' preferences are detailed.

Recommended additions (Addressed):
- Reward Hacking as Safety Risk
  - Status: ADDRESSED
  - Evidence: Section 7

- "Whose Values?" Problem in RLHF
  - Status: ADDRESSED
  - Evidence: Section 7

- Alignment Tax (Capability Reductions)
  - Status: ADDRESSED
  - Evidence: Section 7

- Environmental Impact/Energy Requirements
  - Status: ADDRESSED
  - Evidence: Section 2.4 and Section 7

- Bias Amplification
  - Status: ADDRESSED
  - Evidence: Section 2.3 and Section 7

**Claims And Evidence:**

Yes

**Claims Explanation:**

The paper's descriptive claims about the current state of RL for LLMs are well-supported through extensive citations. Mathematical formulations are rigorous (LLM generation <> MDPs <> state transitions <> reward structures). The discussion of system-level challenges, particularly the training-inference mismatch causing "effective off-policy behavior," is insightful and now includes concrete quantification.

My concern in the initial review was that some algorithmic claims needed more than citations and the revision addresses this satisfactorily. The training-inference mismatch discussion is insightful. The observation that numerical discrepancies effectively convert on-policy to off-policy training isn't widely acknowledged, and quantifying the impact gives readers concrete expectations.

Areas requiring strengthening (Addressed):

~~1. Algorithmic Comparisons~~

~~2. Empirical Evidence~~

~~3. Systems Claims~~

~~From my experience implementing RL controllers, I can confirm the high variance issues in reward signals are indeed critical and accurately characterized. The paper's insight that simpler methods can achieve competitive variance reduction is valuable, though the cost-benefit analysis deserves more empirical ground truth.~~

Strengthening recommendations(Addressed):

~~- Include summary tables comparing algorithms on standardized benchmarks~~

~~- Provide computational cost comparisons (FLOPs, memory, wall-clock time) if available from cited literature~~

~~- Add ablation study results or comparisons when discussing relative algorithm performance~~

~~- Explicitly discuss when PPO may still be preferred despite higher cost~~

**Requested Changes:**

1. Enhance Algorithmic Performance Evidence [Section 3.1]
- Original Request: Add infographics/tables synthesizing empirical results (Algorithm <> Benchmark <> Model Size <> Performance
- Status: ADDRESSED
- Evidence: The authors added Figure 7 comparing PPO, ReMax, GRPO, and RLOO across alignment and math reasoning tasks ie. actual pass@1 vs. pass@256 curves. Table 2's comparison of computational overhead and gradient bias/variance properties is also helpful.

2. Clarify PPO Complexity-Performance Trade-off [Section 3.1]
- Original Request: Frame PPO discussion as cost-benefit analysis rather than theoretical necessity; acknowledge scenarios where PPO's learned baseline may outperform heuristics
- Status: ADDRESSED
- Evidence: Section 3.1 revised from "PPO is overshot for LLMs" to "PPO is computationally expensive for LLMs", appropriately reframing as computational consideration. The revision now acknowledges that token-level baselines from value networks can help with fine-grained credit assignment when well-trained, which was my concern about dismissing PPO too categorically.

3. Expand Process Rewards Discussion [Section 2.2]
- Original Request: Add subsection covering process reward approaches (MATH-SHEPHERD, stepwise rewards), comparison with outcome-based rewards
- Status: ADDRESSED
- Evidence: New dedicated subsection "Process Reward Models: A Comparative Perspective" added to Section 2.2. It covers Math-Shepherd, PRM800K, and explains why current SOTA reasoning models like DeepSeek-R1 still rely on outcome-based rewards despite PRM's theoretical appeal. The discussion of scalability challenges (annotation costs, step-level correctness ambiguity) appears balanced.

4. Add Broader Impact Statement [Section 7]
- Original Request: Create dedicated section addressing reward hacking, "whose values?" problem, alignment tax, energy requirements, bias amplification
- Status: ADDRESSED
- Evidence: New Section 7 "Broader Impact Statement" added, covering: Reward Hacking and Safety, Value Pluralism and Fairness and Capability Trade-offs and Resource Costs. The authors created a dedicated section covering reward hacking, value pluralism ("whose preferences?"), alignment tax, and energy costs. This is woven well with Section 2.3's new paragraph on preference-collection bias.

5. Can Expand Offline Algorithms Coverage [Section 3.2]
- Original Request: Include IPO, KTO, comparative analysis of when to prefer offline vs online methods
- Status: ADDRESSED
- Evidence: New subsection "Variants Beyond DPO" added covering IPO and KTO. The "When to Prefer Offline vs. Online Methods" subsection provides practical guidance: infrastructure compatibility of offline methods, O(m+n) vs. O(m×n) complexity comparison, and when offline methods retain advantages (high coverage datasets).

6. Can Enhance Systems Discussion Quantification [Section 4.3]
- Original Request: Add concrete numbers for GPU utilization, memory savings, training-inference mismatch impact; include reward model MLOps challenges
- Status: ADDRESSED
- Evidence: Section 4.3 significantly enhanced with Quantified training-inference mismatch and Systems overhead quantification. The "Reward Model MLOps and Reward Serving" subsection addresses the operational reality that reward models need versioning, monitoring, and rollback capabilities.

7. Can Improve Future Directions Specificity [Section 5]
- Original Request: Detail specific challenges for multi-modal RL (cross-modal reward alignment, action space design) and tool usage with concrete technical challenges
- Status: ADDRESSED
- Evidence: Section 5 substantially expanded with Tool Usage and Multi-Modal RL. Also, tool usage now has concrete challenges: long-horizon credit assignment across multi-tool trajectories, calibration under API failures, and pointers to benchmarks. Multi-modal RL includes cross-modal reward alignment and action space design questions.

---

> ### Comment · Action_Editor_9dDd · 2026-05-18
> **Official recommendation for the paper.**
>
> Hi reviewer WcTz ,
>
> Would it be possible for you to submit an official recommendation for the paper using the Openreview system (whether you think it should be accepted / rejected etc.).?
>
> Thanks,
> AE

---

### Review · Reviewer_2pbA · 2026-06-03

**Summary Of Contributions:**

This paper provides a survey of reinforcement learning methods for large language models. It first reviews the background of LLMs and RL and formulates language generation as an RL problem. Then it summarizes the main online and offline RL algorithms used for LLM post-training, including policy-gradient based methods, preference optimization methods, and other recent variants. The paper also discusses the large-scale RL training framework from the system perspective, including model updating, response sampling, and system integration. Finally, it provides discussions on future directions such as scaling RL for LLMs, multimodal models, and agentic LLM systems.

**Additional Comments:**

N/A

**Audience:**

Yes

**Audience Explanation:**

Reinforcement learning for large language models is a timely and important topic in machine learning, especially because RLHF, preference optimization, reasoning-oriented RL, and large-scale post-training systems have become central techniques in current LLM development. Researchers working on reinforcement learning, language models, alignment, reasoning, and scalable training systems would likely find this survey useful. The paper is especially helpful for readers who want a structured overview of how classical RL formulations are adapted to the LLM setting.

**Claims And Evidence:**

Yes

**Claims Explanation:**

The paper argues that reinforcement learning has become an important paradigm for improving and aligning large language models, and this claim is well supported by a broad set of references covering RLHF, preference optimization, online RL algorithms, offline algorithms, reasoning models, and large-scale training systems. The paper also provides a clear formulation of LLM generation as an RL problem, which helps connect the classical RL notation with the LLM post-training setting.

**Requested Changes:**

- Adding a discussion comparing this survey with existing surveys on RLHF, preference optimization, and RL for LLMs would be helpful. The current manuscript would be stronger if it clearly explained what unique taxonomy, perspective, or insight it provides compared with prior survey papers.

- Strengthening the critical comparison between different RL algorithms would be helpful. For example, the paper can add tables comparing PPO, DPO, reward-model methods, offline RL methods, and recent reasoning-oriented RL methods in terms of assumptions, reward signal, on-policy/off-policy nature, stability, sample efficiency, and computational cost.

---

> ### Comment · Action_Editor_9dDd · 2026-06-08
> **Official Recommendation request**
>
> Would it be possible for you to submit an official recommendation for the paper?

---

> > ### Comment · Reviewer_2pbA · 2026-06-08
> >
> > Dear Action Editor,
> >
> > I would be happy to submit my recommendation for the paper. However, I am unable to see the submission page or link on my dashboard.
> >
> > Best,
> > Reviewer 2pbA

---

> ### Author Response · Authors · 2026-06-08
>
> We sincerely thank the reviewer for the constructive comments and positive assessment of our survey. We agree that the manuscript would benefit from clearer positioning relative to existing surveys and a more explicit comparison of RL optimization methods for LLM post-training. We have revised the paper accordingly.
>
> **Response to Requested Change 1:** Thank you for this important suggestion. We have added a new paragraph, “Scope and relation to existing surveys,”  in the Introduction (Page 3). This paragraph clarifies how our survey relates to existing surveys on AI alignment, reward design, direct preference optimization, and broader RL-for-LLM overviews. We now emphasize that our survey takes the RL formulation of autoregressive language generation as the organizing principle, highlighting LLM-specific properties such as deterministic token-level transitions, sparse sequence-level feedback, large discrete action spaces, and rollout pipelines whose throughput and staleness affect feasible updates.
>
> We also clarify the unique taxonomy of our survey. Rather than organizing the literature mainly by application domains or method catalogs, we organize the discussion around how the RL formulation shapes problem modeling, optimization design, and training systems. In particular, we emphasize how feedback sources, baseline or value-model designs, online rollout requirements, and system constraints affect the stability, efficiency, and computational cost of RL methods for LLM post-training.
>
> **Response to Requested Change 2:** Thank you for suggesting a more critical comparison across RL algorithms. We have added Table 2 at the beginning of Section 3 (Page 13), before presenting individual algorithms. This table provides a compact comparison of the main optimization classes most relevant to LLM post-training, using four design-relevant dimensions: feedback type, training regime, value-model use, and online rollout use.
>
> This new table also clarifies a key distinction in our survey: RLHF and RLVR are feedback/problem formulations, whereas PPO, TRPO, A3C, REINFORCE, ReMax, GRPO, RLOO, DPO, IPO, and KTO are optimization algorithms or objectives. The surrounding text in Section 3 (Page 13) now explains this distinction and summarizes how these choices affect stability and computational cost.
>
> We also revised the opening of Section 3 (Page 13) and Section 3.2 (Page 21) to avoid confusion with classical offline RL. Finally, we would like to clarify that the “offline” part of our survey focuses on preference-based offline optimization for LLMs, while classical offline RL methods such as BCQ, CQL, and IQL are not treated as a central algorithm family because they have not yet been widely adopted for large-scale LLM post-training. We point readers to existing offline RL surveys for that separate literature.

---

> > ### Comment · Reviewer_2pbA · 2026-06-10
> >
> > Thanks for the detailed responses. They addressed all my concerns.